# Event generation for probabilistic flood risk modelling: multi-site peak flow dependence model vs. weather generator based approach

Benjamin Winter[1,2], Klaus Schneeberger[1,2], Kristian Förster[3], and Sergiy Vorogushyn[4]

[1]Institute of Geography, University of Innsbruck, Innrain 52f, A-6020 Innsbruck, Austria
[2]alpS, Grabenweg 68, A-6020 Innsbruck, Austria
[3]Institute of Hydrology and Water Resources Management, Leibniz University Hannover, Appelstr. 9A, D-30167 Hannover, Germany
[4]GFZ German Research Centre for Geosciences, Hydrology Section, Telegrafenberg, D-14473 Potsdam, Germany

**Correspondence:** B. Winter (Benjamin.Winter@uibk.ac.at)

**Abstract.** Flood risk assessment is an important prerequisite for risk management decisions. To estimate the risk, i.e. the probability of damage, flood damage needs to be either systematically recorded over long period or it needs to be modelled for a series of synthetically generated flood events. Since damage records are typically rare, time series of plausible, spatially coherent event precipitation or peak discharges need to be generated to drive the chain of process models. In the present study, synthetic flood events are generated by two different approaches to model flood risk in a meso-scale alpine study area (Vorarlberg, Austria). The first approach is based on the semi-conditional multi-variate dependence model applied to discharge series. The second approach relies on the continuous hydrological modelling of synthetic meteorological fields generated by a multi-site weather generator and using an hourly disaggregation scheme. The results of the two approaches are compared in terms of simulated spatial patterns of peak discharges and overall flood risk estimates. It could be demonstrated that both methods are valid approaches for risk assessment with specific advantages and disadvantages. Both methods are superior to the traditional assumption of a uniform return period, where risk is computed by assuming a homogeneous return period (e.g. 100-year flood) across the entire study area.

## 1 Introduction

In recent decades several large flood events occured across Europe resulting in direct damage exceeding one billion Euro (Kundzewicz et al., 2013). Growing flood damage due to socio-economic and land-use changes as well as a possible increase of flood hazards in a warmer climate (IPCC, 2018) calls for robust flood risk assessment. A reliable estimation of flood damage is an essential prerequisite for profound decision making (de Moel et al., 2015). The most straightforward estimation of possible flood risk would be a statistical evaluation of documented flood damage across the area of interest. In practice, systematic damage records are rare and mostly not available for longer periods (Downton and Pilke, 2005), whereas the major interest e.g. in the re-insurance industry is on losses due to extreme events such as the 200 years return period to fulfill the Solvency II regulations. (European Union, 2009).

Following the European flood directive flood risk is defined as "the combination of the probability of a flood event and of the potential adverse consequences [...]" (European Union, 2007). In other words, flood risk is defined by the probability of damage. Hence, for risk estimation, a flood event including its probability of occurrence (hazard) on the one hand and the vulnerability of exposed values on the other hand need to be considered (Klijn et al., 2015). Since risk assessment is currently not feasible based on empirical data, modelling approaches based on synthetic flood scenarios are often deployed (e.g. Lamb et al., 2010; Falter et al., 2015; Schneeberger et al., 2019).

In a traditional approach, the hydrological load is estimated by means of extreme value statistics using river gauge data and transformed into corresponding inundated areas by hydrodynamic models (Teng et al., 2017). The monetary damage can then be assessed in combination with susceptibility functions which describe the relationship between one or more flood hazard characteristics (e.g. inundation depth, flow velocity) and damage for the elements at risk (Merz et al., 2010). This approach implies two strong assumptions. First, the return period of flood discharge is assumed to be equal to the return period of the resulting damage. Second, a uniform return period across the entire study area is considered and resulting damage estimates are accumulated. The first assumption can be relaxed by modelling a continuous series of synthetic flood events. As a result, a long series of damage values can be generated and used for analysing damage frequency distribution (Achleitner et al., 2016). The second assumption of homogeneous flood return periods may be valid for small areas (de Moel et al., 2015). With increasing scale, the assumption of a homogeneous return period becomes unlikely, as precipitation and flood footprints are inhomogeneous in space. This assumption can lead to an overestimation of risk for specific return periods in large river basins (Thieken et al., 2015; Vorogushyn et al., 2018; Metin et al., 2020). To overcome the second limitation, realistic spatially heterogeneous events need to be generated across the area of interest which fully represent the spatial variability of flooding (Schneeberger et al., 2019).

Generation of spatially heterogeneous flood events in terms of precipitation fields or discharges is of current scientific interest (Keef et al., 2013; Falter et al., 2015; Falter, 2016; de Moel et al., 2015; Speight et al., 2017; Diederen et al., 2019; Diederen and Liu, 2019; Schneeberger et al., 2019). There are different approaches to generate large event series of heterogeneous flood events. One possibility is the application of multivariate statistical methods to discharge series, such as copula models (Jongman et al., 2014; Serinaldi and Kilsby, 2017; Brunner et al., 2019) or the semi-parametric conditional model proposed by Heffernan and Tawn (2004) (hereinafter referred to as 'HT-model'). These models consider the pairwise dependence of peak discharges at multiple locations and generate synthetic series of multiple dependent flow peaks. The second possibility is based on the generation of spatially distributed meteorological fields by a weather generator, either station-based with subsequent interpolation (Falter et al., 2016; Falter, 2016; Breinl et al., 2017; Evin et al., 2018; Raynaud et al., 2019) or raster-based (Buishand and Brandsma, 2001; Peleg et al., 2017). Synthetic meteorological fields are subsequently used to drive hydrological simulations to generate streamflow values across the study area.

The two presented approaches estimate the hydrological load in the river network at multiple locations, but are different in their nature. This leads to the key question of the present study: Does it matter which approach is chosen in the context of flood risk modelling, and what are advantages and disadvantages of the two? We answer this question by comparing the set of heterogeneous flood events from the HT-model with the one resulting from a weather generator and subsequent rainfall-runoff

modelling. Both methods are embedded in a probabilistic flood risk model used to estimate the effect of chosen methods on flood losses. To the authors' best knowledge, there is no study to date in which the two approaches are directly compared. Additionally, the flood risk corresponding to homogeneous flood scenarios of certain return periods ("traditional" approach) is derived and compared to the other two approaches.

This paper is organised as follows: First, the study area is shortly described. In the second section the flood risk model is introduced and the two different approaches for heterogeneous event generation are presented in details. Section three presents the results of the comparison, which are discussed in the following section. Finally, conclusions summarise the major findings.

## 2    Study Area

The flood risk model is applied in the westernmost province of Austria, Vorarlberg. The region is characterised by a strong
altitudinal gradient between the Rhine Valley ($\approx 400$ m a.s.l.) and the high mountain ranges of the Alps ($> 3000$ m a.s.l.). As a result of the high relief energy, the rivers are characterised by a fast hydrological response with short concentration times. The mountainous landscape of in total 2600 km$^2$ is dominated by forest, meadows and pastures with only small percentage of settlement area (Sauter et al., 2019). Due to steep topography, asset values are concentrated in the lowlands of larger valley floors, especially alongside the Rhine and Ill rivers. Vorarlberg is characterised by one of the highest precipitation amounts in
Austria conditioned by predominantly westerly flows and strong orographic effects (BMLFUW, 2007). During the last decades, the province was affected by several severe flood events in 1999, 2002, 2005 and 2013. The most devastating recent flood event in August 2005 caused about €180 million direct tangible losses for the private and public sector, including infrastructures (Habersack and Krapesch, 2006). Figure 1 provides an overview of the study area, including the river network, settlement areas and the locations of river gauging stations as well as meteorological stations.

## 20    3    Methods and Data

The probabilistic flood risk model (PRAMo) used in the presented work consists of three different modules: The Hazard module comprising the generation of long time series of flood events, the Vulnerability module used to evaluate possible adverse consequences of flood events with a certain exceedance probability, and the Risk Assessment module which combines the results of the Hazard and Vulnerability modules to estimate the loss per event and resulting risk (Schneeberger et al., 2019).
The output of the flood risk model are expected annual damage and exceedance probability curves of damage. PRAMo was previously driven by the synthetic flood event series of coherent peak discharges generated by the HT-model (Schneeberger and Steinberger, 2018). A second event generation approach based on a multi-site, multi-variate weather generator and continuous rainfall-runoff modelling was recently introduced by Winter et al. (2019) and is used for comparison with the HT-model based approach and the assumption of homogeneous return periods. Figure 2 provides an overview of the modules and the simulation
steps, which are described in more details in the following.

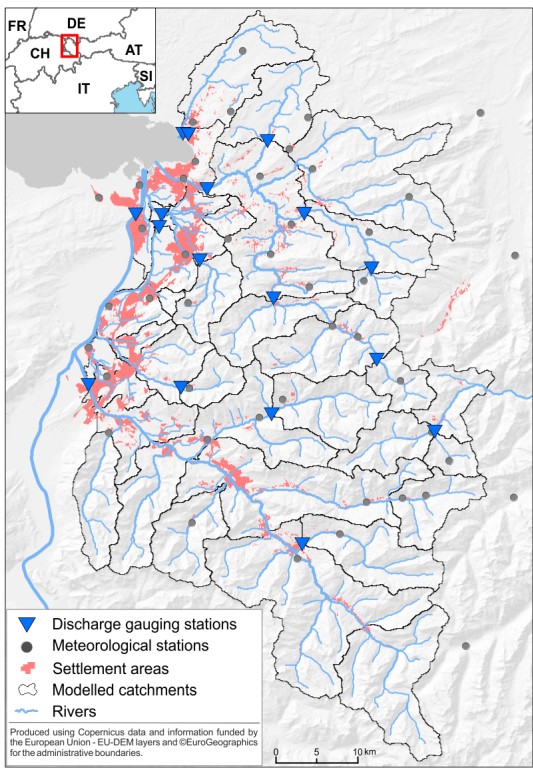

**Figure 1.** Study area and the location of meteorological and river gauging stations.

In this study, data of 17 gauging stations (1971-2013) are applied for the HT-approach. The continuous simulation of the WeGen approach is based on daily time series from 1971-2013 for 45 meteorological stations (c.f. Figure 1). At hourly time steps data for only 23 sites starting from 2001 are available. Stations without hourly information were interpolated by an inverse distance-weighting scheme (for details see Winter et al. 2019).

## 3.1 Hazard Module I: HT-model

The Hazard module generates time series of spatially distributed synthetic flood events. In the first approach, we apply the conditional extreme value model (HT-model) proposed by Heffernan and Tawn (2004) to peak flows. In this approach, flood events are understood as a set of spatially consistent peak discharges at multiple locations of stream gauges. Spatial consistency is ensured by considering the correlation structure of peak flows from the past observation period. Discharge time series at 17 gauges across the study area are used to parameterise the HT-model. In the first step, the observed data are standardised by a marginal model to a Laplace distribution. In the second step, the dependency between the stations is modelled for the case that peak flow at one station is above a certain threshold. According to Lamb et al. (2010), the HT-model can be interpreted as a multi-site peak-over-threshold approach. Due to strong seasonality of streamflow in Vorarlberg, the HT-model is separately parameterised for winter and summer periods (Schneeberger and Steinberger, 2018).

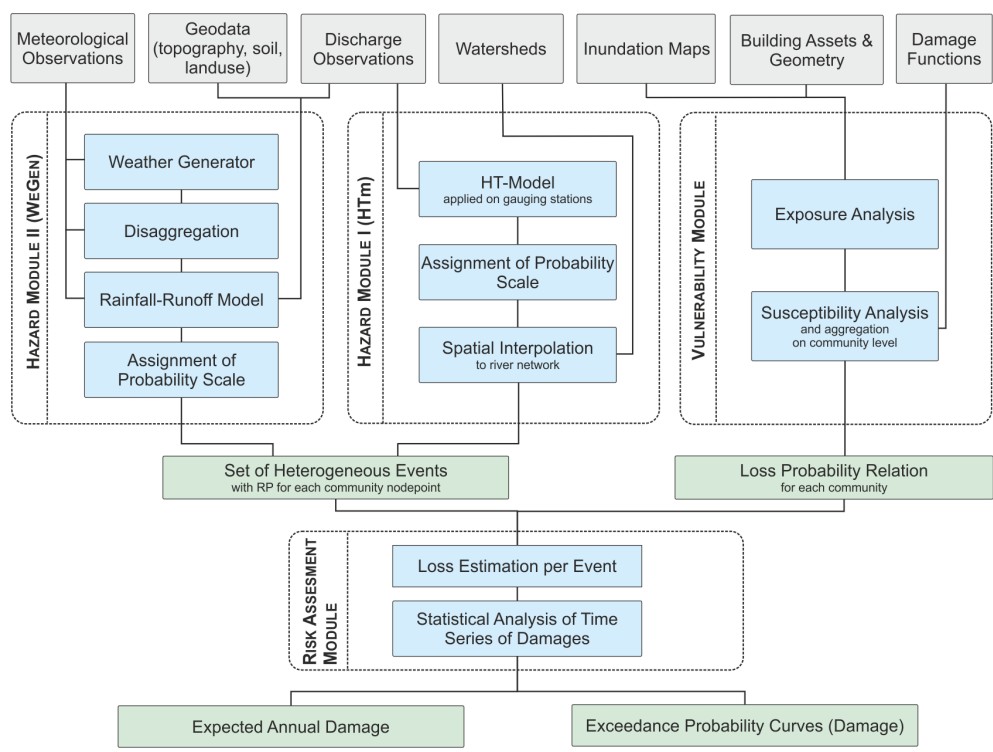

**Figure 2.** Flowchart of the PRAMo flood risk model including two different approaches for flood event generation.

For the set of synthetic flood peaks at each of the 17 gauge locations we estimate the return period based on the Generalized Extreme Value (GEV). A flood event is characterised by exceedance of a certain streamflow at a single or multiple location with a defined time period. As threshold for defining a widespread flood event, a return period of 30 years was selected in the present study. The output of the HT-model in terms of synthetic flood peaks is available at the locations of gauging stations.

5  Hence, for the river segments without observations, the flows and their respective return periods need to be estimated. We apply the top-kriging approach (Skøien et al., 2006) for the spatial interpolation of model results to the entire river network. This method takes into account the nested structure of river catchments which makes the results more robust compared to traditional regional regression based approaches (Laaha et al., 2014; Archfield et al., 2013). A more detailed description of the HT-model is provided in Schneeberger and Steinberger (2018) and Schneeberger et al. (2019).

10  ## 3.2  Hazard Module II: WeGen

The second approach is based on a stochastic weather generator used to drive a hydrological model. Long-term daily precipitation and temperature series are generated, with a multi-site, multi-variate weather generator based on the auto-regressive model (Hundecha et al., 2009). Daily precipitation amounts are generated from mixed gamma and generalised Pareto distributions fitted to individual weather stations. The mixed distribution is shown to better capture extreme precipitation while robustly

modelling the bulk of precipitation amounts (Vrac and Naveau, 2007). In respect to seasonal patterns, the fitting is applied on a monthly base. Occurrence and amount of precipitation are modelled considering the autocorrelation and inter-site correlation structure. The mean temperature is then modelled conditioned to the simulated precipitation (Hundecha and Merz, 2012). As the study area is characterised by mostly alpine topography with short catchment response times, the hydrological model

needs to be driven by meteorological input at sub-daily resolution to estimate realistic peak flows (e.g. Dastorani et al., 2013). A non-parametric k-nearest neighbour algorithm based on the method of fragments is applied to disaggregate the generated daily values to hourly time steps (Winter et al., 2019). For a day to disaggregate, the generated daily values of temperature and precipitation from the weather generator are compared against observed daily data at all stations. Subsequently, k-nearest neighbours in terms of lowest euclidean distances between generated and observed daily values are selected. Next, one match-

ing day is randomly sampled from the selected neighbours and the corresponding relative temporal patterns from the match day are transferred to the input day (method of fragments). In contrast to the previous study (Winter et al., 2019), a centred moving window of 30 days is applied instead of the identical months in order to restrict the search of possible matching days. The modification increases the variability between the disaggregated days and reduces the maximum search distance on a temporal scale, especially for days at the beginning and end of a month.

Following the generation of meteorological data at the locations of the weather stations, a spatial interpolation to continuous meteorological fields is necessary for the application of the rainfall-runoff model. Complex methods for spatial interpolation can be applied (e.g. Goovaerts, 2000; Plouffe et al., 2015), however, for the long term simulation a computationally efficient approach is needed. The interpolation was carried out by a inverse distance-weighting scheme including a step wise lapse rate to account for the complex topography (Bavay and Egger, 2014).

Finally, the semi-distributed conceptual rainfall-runoff model HQsim is applied to simulate streamflow across all catchments of the study area (Kleindienst, 1996). HQsim is forced by precipitation and temperature data and was previously used in various studies in alpine catchment areas (e.g. Senfter et al., 2009; Achleitner et al., 2012; Dobler and Pappenberger, 2013; Bellinger, 2015; Winter et al., 2019). A simulated annealing algorithm is used for the model calibration against observed discharge data at the gauging stations (Andrieu et al., 2003). From a long synthetic discharge series, relevant flood events are identified and

extracted. For this, a flood frequency analysis at all points of interest based on fitting the GEV-distribution using the L-moments is carried out. Analogously to the HT-model approach, a threshold of 30 year return period, at least at one site across the study area is applied to define relevant flood events. A more detailed description of the modelling chain, including the disaggregation procedure, is given in Winter et al. (2019).

### 3.3   Vulnerability Module

While the Hazard module computes the hydrological load, the Vulnerability module assesses the possible negative consequences in terms of exposed objects and monetary damage. The module is based on the widely used approach of combining the exposure and susceptibility of elements at risk in the inundated areas (Koivumäki et al., 2010; Merz et al., 2010; Huttenlau and Stötter, 2011; Meyer et al., 2013; Cammerer et al., 2013; de Moel et al., 2015; Falter, 2016; Wagenaar et al., 2016). The module calculates losses for each community in the study area for a number of predefined return periods (or probabilities) (i.e.

RP = 30, 50, 100, 200 and 300 years). The results of the Vulnerability module are loss-probability relations for each community, describing the expected damage for the corresponding return periods. To derive a continuous relation, a linear interpolation between available data points (RP-damage) is applied. The loss-probability relations are used as input in the Risk Assessment Module and combined with the simulated return periods (Hazard Module) at each community to derive risk curves.

5      At the scale of a community (in average 28 $km^2$), a homogeneous return period of hydrological load is assumed and associated with the total community loss. For the loss calculation we use "official" inundation maps. The inundation maps are based on 1D hydrodynamic modelling in rural areas and 2D modelling in urban areas (IAWG, 2010). The boundary conditions for the hydrodynamic simulation are taken from the Austrian flood risk zoning project HORA (Merz et al., 2008).

     The estimation of monetary damage for the elements at risk is based on the relative damage functions combined with the total asset values. A damage function describes the relative loss of value as a function of water depth (Merz et al., 2010). If available, additional damage influencing parameters, such as flow velocity or contamination can be considered for damage assessment (Merz et al., 2013). In accordance to Schneeberger et al. (2019), the one parametric damage model of Borter (1999) is applied in the present study. The damage model was derived for Switzerland, which is a direct neighbour to the Austrian province Vorarlberg with a similar topography and building structure. More precise, site-specific damage functions are not available for the study region.

     The damage estimation is conducted on a single object basis for residential buildings only. To derive the flood losses, the available inundation maps are combined with the asset datasets and damage function. Subsequently, the object based loss data are aggregated for each community. The absolute building values indexed to 2013 according to the construction price index (Statistik Austria, 2019) are derived by calculating mean cubature values from local insurance data, and transferred to the entire building stock of the study area (Huttenlau et al., 2015). Since derived values are based on insurance data, they are consequently defined as replacement values.

## 3.4    Risk Assessment Module

The risk assessment module brings together the results of the hazard and vulnerability modules to generate a time series of losses and calculates the resulting risk curve for the area of interest (Schneeberger et al., 2019). In order to combine the results, each spatial unit (community) is represented by a defined model node point at the river network. For each generated heterogeneous flood scenario, the recurrence intervals are derived for all model node points (hazard module) and combined with the respective loss-probability relation to compute losses (vulnerability module). By integrating the losses at all model node points, i.e. for each community, the total loss for every generated event can be calculated. By evaluating the overall modelled time series of events a continuous time series of damage is generated. Finally, the time series of damage can be statistically analysed to derive the expected annual damage (EAD) and to construct risk curves (Schneeberger et al., 2019). More detailed information about the Vulnerability and Risk Assessment Module, including a schematic overview of the module interaction is provided in Schneeberger et al. (2019).

## 3.5 Assessment of spatial coherence of generated events

A core element of the probabilistic flood risk model is the generation of plausible, spatially heterogeneous flood events. To investigate the spatial coherence of synthetic events generated by two different approaches, two spatial dependence measures proposed by Keef et al. (2009) are applied. The first measure $P_{i,j}(p)$ describes the probability that a dependent site $i$ exceeds a certain threshold, given that a conditional site $j$ is exceeding a threshold $q_p(Q_j)$ as well:

$$P_{i,j}(p) = Pr(Q_i > q_p(Q_i)|Q_j > q_p(Q_j)), \tag{1}$$

where $(p)$ is the level of extremeness (quantile) and $Q_i$ and $Q_j$ are the dependent and conditioned runoff series, respectively. The calculation of the thresholds are based on a three day block maxima, which was found to be appropriate in this region (Schneeberger and Steinberger, 2018). The second spatial dependence measure $N_j(p)$, is an overall summary metric and describes the average probability of all dependent sites $i$ to be high, given that the conditional site $j$ is high, defined as:

$$N_j(p) = \frac{\sum_{i \neq j} Pr(Q_i > q_p(Q_i)|Q_j > q_p(Q_j))}{n-1} \tag{2}$$

In case of the WeGen approach the dependence matrices were computed for the peak discharges at the gauging station locations resulting from the combined simulations of the weather generator and rainfall-runoff model.

## 4 Results

### 4.1 Simulation Results of the Continuous Modelling Approach (WeGen)

To assess the performance of the continuous modelling approach extreme precipitation of simulated data are compared to observed station data (daily: 1971–2013; hourly 2001–2013) for the weather generator and disaggregation procedure. The median and the uncertainty range represented by the 5% and 95% quantiles of 100 model realizations are compared to the observed data. Figure 3a shows the results for 99% quantile of daily precipitation (wet days) for all 45 station and spring (MAR-APR-MAY), summer (JUN-JUL-AUG) and autumn (SEP-OCT-NOV). In general, the characteristics of the observed daily precipitation is well reproduced by the weather generator. A few stations, however, show a slight underestimation in summer (mainly June and August). The validation results for all months separately including maximum and minimum simulated daily temperatures are provided by Winter et al. (2019). To validate the disaggregation procedure, the hourly data are first aggregated to daily data and subsequently disaggregated back to hourly time steps. For the comparison of disaggregated precipitation, 99%, 99.9% and 99.95% quantiles are calculated and compared to the observed values. The results for the 99.9% quantile show a good agreement between observed and simulated precipitation intensities for the three analysed rainfall durations 1, 3 and 6 hours (Figure 3b). Results for the 99% and 99.95% quantile are shown in Winter et al. (2019).

The rainfall-runoff model is calibrated (2001-2007) and validated (2008-2013) in classical split-sample approach (Klemeš, 1986) for all catchments of the study area against observed river gauging data. On average, a Nash-Sutcliffe efficiency (NSE;

Nash and Sutcliffe, 1970) of 0.68 and 0.67 and a Kling-Gupta efficiency (KGE; Kling et al., 2012) of 0.75 and 0.74 are achieved for the calibration and validation periods, respectively. Detailed results for the individual catchments, including a comparison of design flood estimates with a flood frequency analysis and a design storm approach are given in Winter et al. (2019).

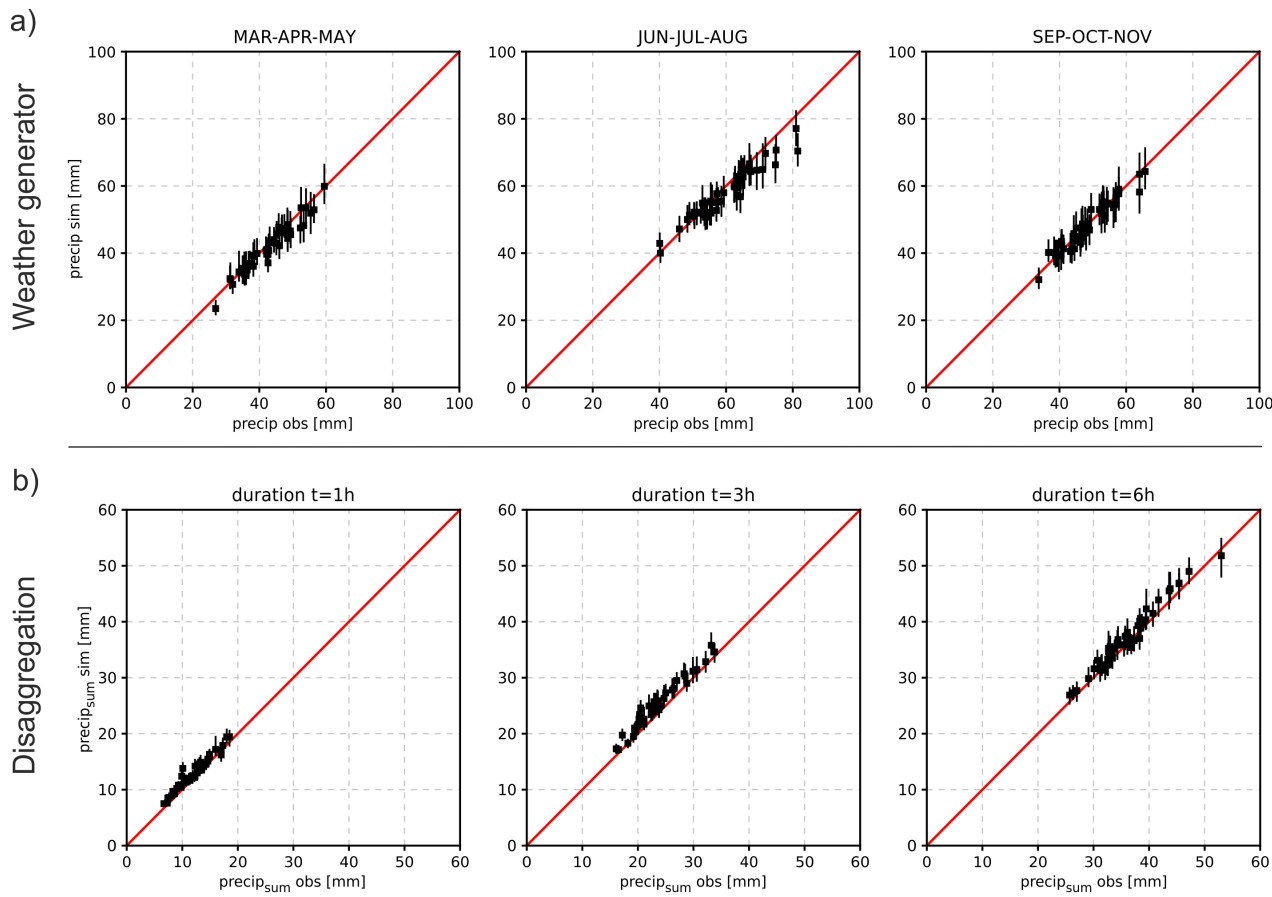

**Figure 3.** Validation results of the weather generator and the revised disaggregation procedure for all stations ($n = 45$). The bars represent the median and the 5 to 95% quantile range of 100 realizations for the weather generator and disaggregation. a) Weather generator: 99% quantile of daily precipitation for generated data compared with observed data for spring, summer and autumn. b) Disaggregation: 99.9% quantile of 13 years of disaggregated data is compared to observed data, for the precipitation sum of 1, 3 and 6 h duration.

## 4.2 Spatial Patterns of Generated Flood Events

For the analysis of spatial coherence, 100 simulations using each of the two event generation approaches (HT-model and WeGen) were carried out. Each simulation comprised 42 years of data corresponding to the length of the observed discharge series. Figure 4 illustrates exemplary results for four gauging stations and both methods. Each plot shows the dependence measure between the two stations depicted on the maps in the principal diagonal. The gauge Kennelbach and Gisingen are

the two largest catchments of the study area (about 80% of the total area). The examples Schruns and Thal are subcatchments of Gisingen and Kennelbach, respectively and thus represent two strongly related gauge pairs. The measure is calculated for discharge values with exceedance probability between $p = 0.99$ and $p = 0.997$, above which the data are too few $(n < 15)$ to calculate a meaningful $P_{i,j}$ value. Based on the empirical distribution function of the 3-day block maxima series, a $p$-value of

5  0.99 refers to a return period of approximately 1 year and a p-value of 0.997 refers to a return period of roughly 3 years.

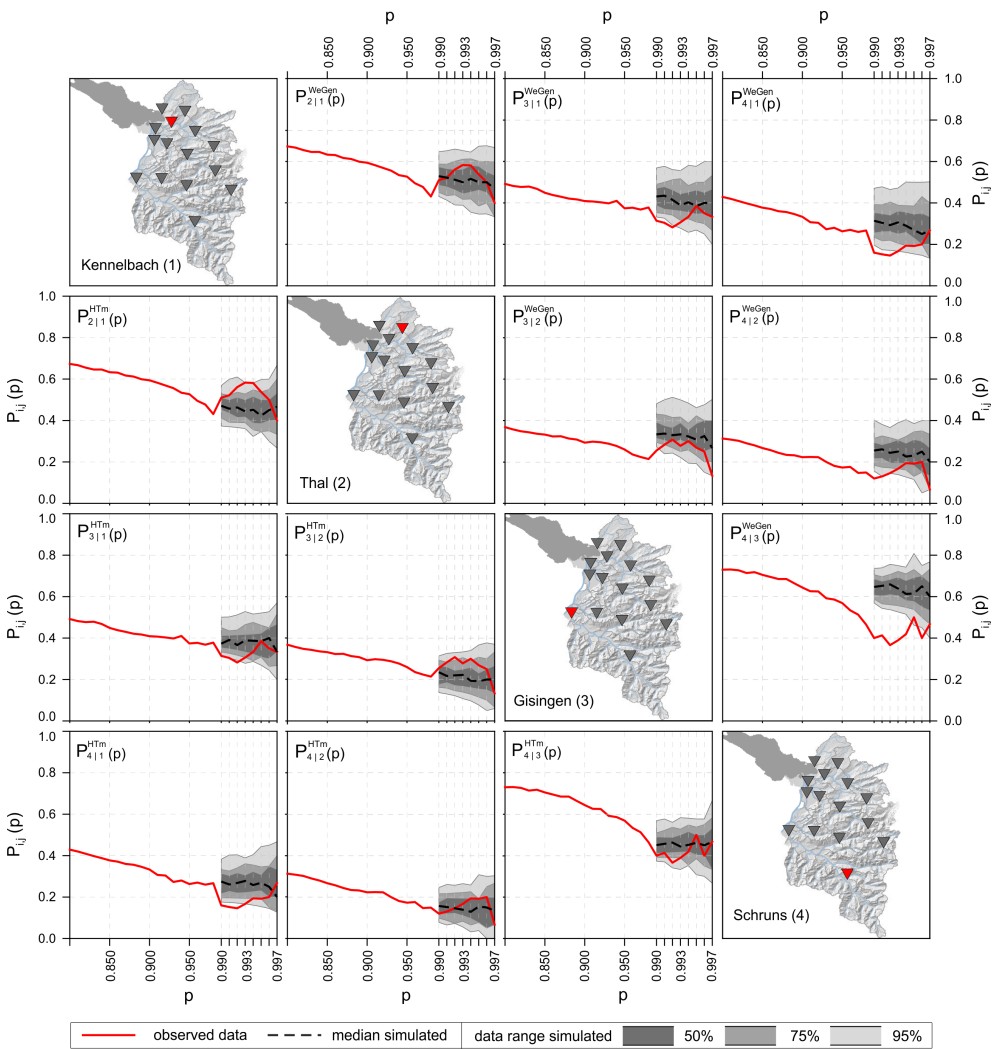

**Figure 4.** Comparison of observed (42 years) and simulated conditioned exceedance probability $P_{i,j}(p)$. The range of the simulated results is based on 42 years of simulation with 100 realizations. The plots in the lower triangle correspond to the HT model, whereas those in the upper triangle show the WeGen results.

In general, the spatial dependence declines with the level of extremeness. For more extreme runoff situations, the dependence structure is less stable and prone to a large variability. The HT-model results in the lower triangle reproduce the observed spatial

patterns between the stations well. The observed measure is in $\approx 90\%$ of the cases inside the simulated data range $(2.5 - 97.5\%$ quantile). The results of the WeGen approach follow the general observed patterns of lower dependence (e.g. $P_{i,j}(p) \approx 0.2$ for Thal (2) vs. Schruns (4)) and higher dependence (e.g. $P_{i,j}(p) \approx 0.5$ Kennelbach (1) vs. Thal (2)). However, the results are biased towards a higher dependence, such that only half of the results correspond well to the observed data.

To analyse the dependence structure of high flows across the study area, the measure $N_j(p)$ is calculated for all node points corresponding to different communities. The measure is calculated for $p$ values corresponding to the 1-year, 10-year and 100-year return period. As the simulation of the two approaches are not limited to the length of the observed data, the results are based on the median of 30 realizations of 1000 years of HT-model and WeGen simulations (Figure 5). The length is chosen to be far above the highest return period of available homogeneous inundation data (RP300) and the number of 30 realizations is
dictated by the computational limitations of the continuous simulation on an hourly time step. Both approaches (see Figure 5 a-c) show a decline of spatial dependence towards higher return periods. The general patterns of lower spatial dependence in the southern part of the study area and of the individual northern catchments is visible. The node points downstream are characterised by a higher dependence. For high return period of 100 years (Figure 5 c), the simulated spatial dependence is higher for the HT-model than for the WeGen results in contrast to the findings for the lower return periods. The results are
regionally different. Whereas the dependence measure is higher for the HT-model in the western part of the study area, the north-eastern catchments show higher degree of dependence for the WeGen approach.

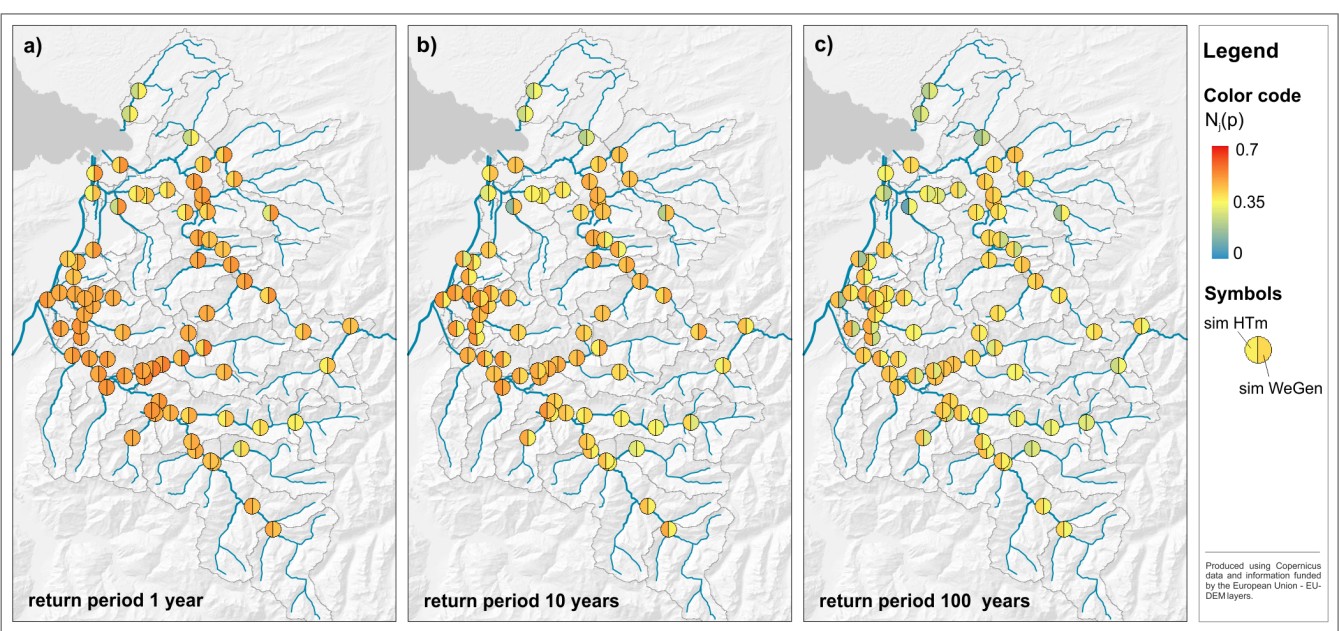

**Figure 5.** Spatial dependence measure $N_j(p)$ for the community node points at the river network and three different return periods. The results show the median for the HT-model and WeGen approach based on 30 realizations of 1000 years simulation.

### 4.3 Comparison of risk curves

To compare the effect of the two approaches of synthetic event generation on the overall estimated loss, flood risk curves are calculated. Confidence intervals are derived based on 30 realizations of 1000-year simulations. Furthermore, the risk curve based on the assumption of homogeneous return period floods across all catchments is derived based on 5 inundation maps corresponding to the return periods between 30 and 300 years. The two synthetic event generators result in a comparable range of overall estimated flood risk (see Figure 6). The WeGen approach systematically overestimates the risk computed by the HT-model. The relative difference between the estimated median values (($WeGen - HTm)/WeGen$) is approximately 17.5%. The uncertainty increases with increasing return period of damage alongside the extrapolation of the input time series. On average 172 damage events are generated per 1000 years of simulation in the WeGen approach compared to about 167 for the HT-model. Both approaches show a significant lower damage in comparison to the assumption of homogeneous scenarios for specific return periods. The estimated damage of a homogeneous 100-year flood scenario is ≈50% above the HT-model results and still 40% above the WeGen approach.

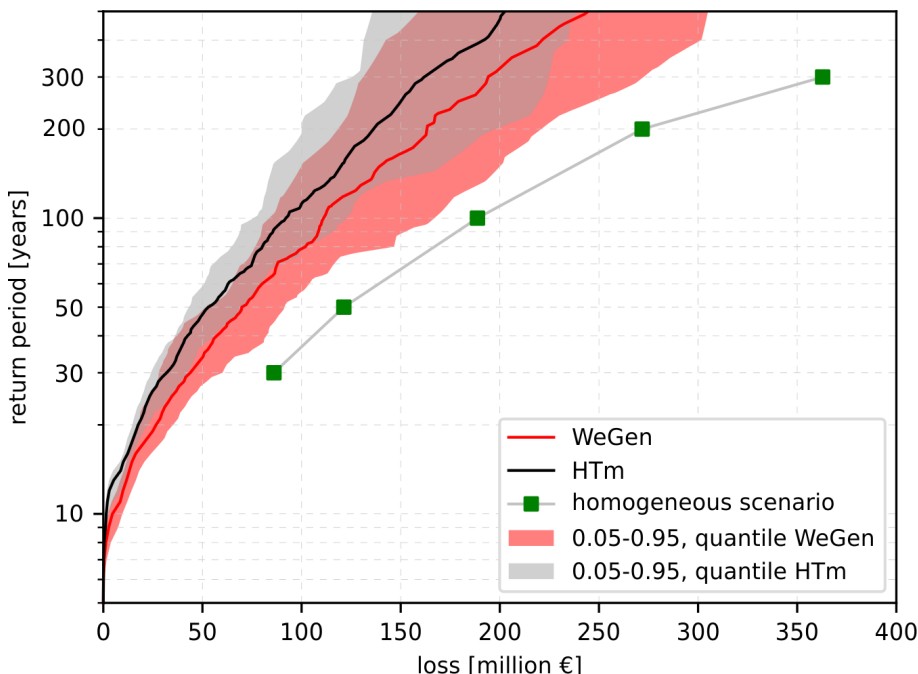

**Figure 6.** Risk curves for WeGen and HT-model approach in comparison to the results of a homogeneous scenario. The median and quantile confidence intervals are based on 30 realizations of 1000 years of simulation. Monetary values are normalized to the year 2013.

The sets of generated heterogeneous flood events reflect a large variability of plausible spatial patterns. Hence the estimated flood risk is the result of a combination of these patterns. Figure 7 shows multiple examples of generated flood events corresponding to an estimated damage of 100±1 million Euro for both model approaches. The general severity in terms of flood

hazard (without consideration of flood risk) is given by the Unit of Flood Hazard (UoFH). The measure UoFH is a simple proxy of hazard severity defined as the total number of sites at which the threshold of 30 years return period is exceeded (Schneeberger et al., 2019). Even though, the selected severity of displayed flood event is rather high, some of the generated events are still spatially limited. The event with the lowest UoFH of 46 corresponds to $\approx 50\%$ of all sites exceeding the 30-year threshold.

The most widespread event (UoFH=77) corresponds to about $90\%$ of the sites exceeding the threshold. This result reflects the spatial distributions of elements at risk with a settlement concentration alongside the larger valley areas in the study area (c.f. Figure 1). Thus, the damage corresponding to an event is largely influenced by the region affected. If the overall comparison is conducted on hazard level only, the impact of widespread flood events may be overestimated, while the impact of spatially limited events in densely populated areas are underestimated.

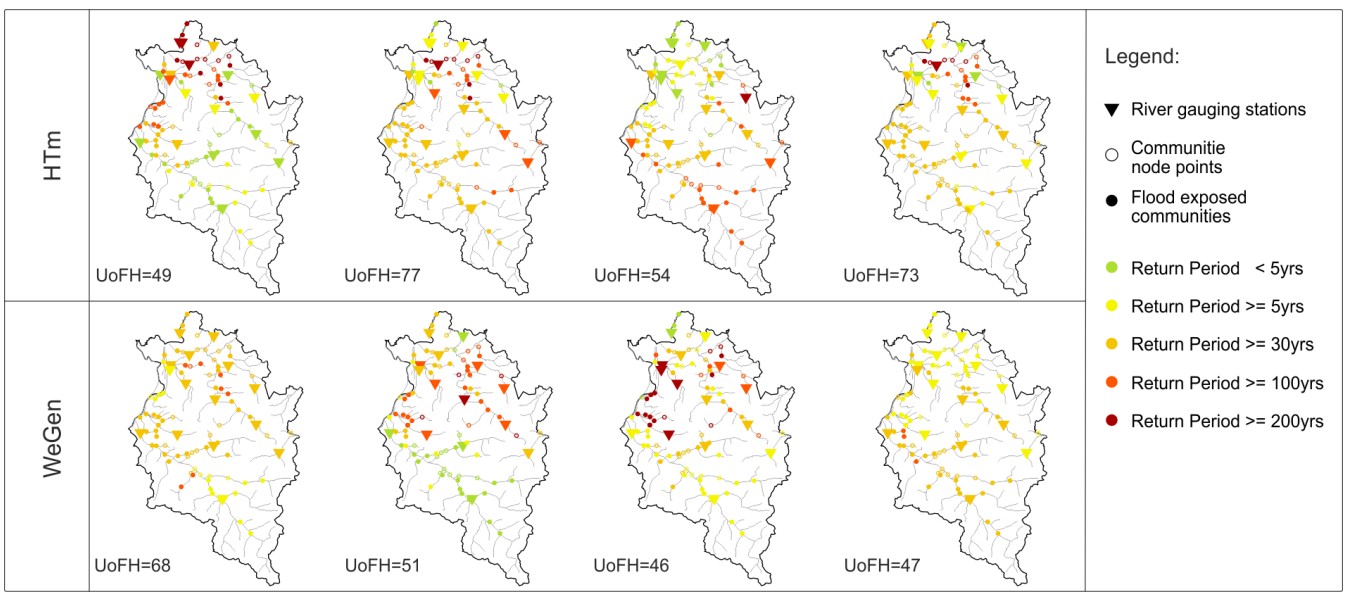

**Figure 7.** Examples of flood events with an estimated flood damage of 100±1 million Euro flood damage for HT-model and WeGen approach. The general severity of flood events is characterised by the Unit of Flood Hazard (UoFH).

## 5   Discussion

Both approaches, the HT-model and the WeGen approach, simulate complex spatially heterogeneous patterns of synthetic flood events. In the present study, the HT-model outperforms the WeGen approach in terms of reproducing the observed dependence patterns of peak flows at the gauging stations. The HT-model makes use of the observed river gauging data and models their dependence structure directly. On the contrary, the WeGen approach models the dependence structure only indirectly based on

the meteorological input data.

The overall river network and especially small ungauged tributaries do however rely on the top-kriging interpolation in case of the HT-model approach and are not able to react independently to the larger river system. This explains the higher dependence structure on the community node points, while at the river gauges the results do correspond well to the observed values. Nevertheless, in both cases the capability to capture spatial effects of a certain spatial scale in the end depends on the density of the measuring network and its data quality.

The WeGen approach seems to overestimate the overall spatial dependence in the study area in comparison to the observed values. This was also found in a previous study, comparing a different set of gauging stations (Winter et al., 2019). One possible reason could be, the spatial interpolation of the meteorological data by the rather simple IDW-approach, without consideration of shading effects etc. and the rather short length of hourly input data for the disaggregation procedure might affect the spatial patterns towards a stronger dependence. More importantly, the WeGen model itself seems to overestimate the dependence between stations particularly for higher return period thresholds. This is in line with the results of the recent evaluation of the weather generator (Ullrich et al., 2019), which suggest an overestimation of correlation of extreme precipitation between individual stations leading to an overestimation of areal rainfall. The correlation structure of the weather generator is fitted on a monthly base, independently of the rainfall intensities and thus does mix low intensity large scale rainfalls and small scale convective events. The simulated stronger spatial dependence in certain areas with high damage potential also contributes to the higher flood risk estimate by the WeGen approach.

Only one possible combination of weather generator, disaggregation procedure and rainfall runoff model was applied for the WeGen approach. Thus, by the application of an alternative weather generator with different assumptions about the spatial dependence or tail distribution, the resulting risk estimates may change. This counts as well for the application of an different rainfall runoff model or alternative disaggregation procedure (e.g. Müller-Thomy et al., 2018). Thus, the result of a higher risk estimate for the WeGen approach in comparison to the HT approach can not be generalized to other model combinations.

Both approaches for synthetic event generation differ substantially in terms of estimated damage from the one assuming a uniform return period across the whole study area (Figure 6). The flood losses for individual return periods above the 30-year threshold under the homogeneous assumption are largely overestimated. This result confirms the necessity to take heterogeneous spatial patterns into account. An event where every community in the study area is affected by discharges exceeding the 30-year return period during a single event is rare. Based on total of 30000 years of simulation less then 10% of the communities experience losses simultaneously in more than 50% of events (Figure 8). It can be expected that with increasing spatial scale, the likelihood that a large number of communities will experience high return period discharges and losses in a single event will decrease (Metin et al., 2020). Therefore, generation of spatially consistent heterogeneous flood events is particularly important with increasing spatial scale. At the same time, considering dependence of meteorological and hydrological variables at multiple locations with increasing scale and increasing number of dependent locations becomes more challenging.

A fundamental difference between the two approaches resides in the way of considering the hydrological processes. The HT-model takes a purely statistical approach by analysing the dependence of peak discharges above a certain threshold. It does not explicitly consider hydrological processes which generate extremes. For instance, the non-linearity of catchment response is not explicitly taken into account, but only so far it is imprinted in the previously observed peaks used for model parameterisation.

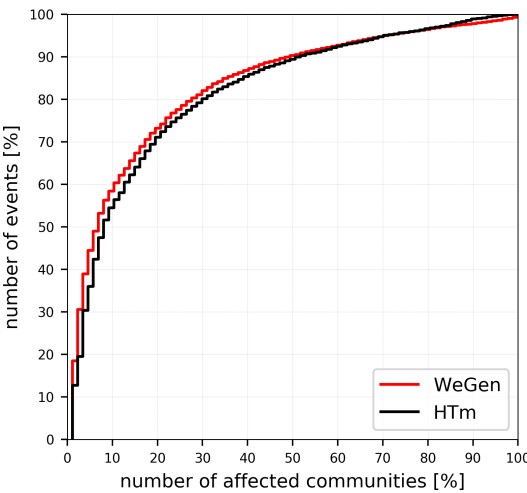

**Figure 8.** Relative number of flood events exceeding a 30-year flood threshold and corresponding relative number of affected communities. The results are based on 30000 years of simulation.

The combination of the weather generator and rainfall-runoff modelling describes the hydrological processes in a spatially consistent and time continuous way. Hence, the effect of soil moisture accumulation and pre-event catchment conditions are explicitly modelled. By the application of a fully distributed, physically based model the hydrological process description could even be improved for example by solving full energy balance equations for snow melt or evapotranspiration (e.g. Förster et al.,
2014, 2018). On the downside, a further increase in model complexity might compromise the model parameter identifiability, increase calibration effort and computational burden and increase input data demand (temperature, precipitation, radiation, humidity and wind speed).

In general, continuous hydrological modelling generates full hydrographs at all locations that allows for direct coupling with hydraulic models as for example applied in Falter et al. (2015) and Falter (2016). The direct coupling of the WeGen approach
with a 1D-2D hydrodynamic model would also allow to consider hydrodynamic interactions in the river network and their possible affect on the risk estimates. This may for example be the reduction of risk downstream due to dike overtopping and failure upstream. In case of the HT-model, only peak discharge of events is estimated, not the entire hydrograph. Hence, these results cannot be used directly as a boundary condition for unsteady hydraulic simulations. Assumptions on the shape of a hydrograph would be required.
In addition, the continuous modelling approach is capable to explicitly model scenarios of changing hydrological boundary conditions. For instance, changes in the climate system can be taken into account in the generation of meteorological fields by conditioning the rainfall and temperature probability distributions (e.g. Hundecha and Merz, 2012). Also possible changes in land use can be considered by parameterising hydrological models accordingly (Rogger et al., 2017). As the HT-model approach is based on observed streamflow only, change scenarios may be included in terms of trends. However, they cannot

be modelled explicitly. A continuous simulation approach requires a vast amount of processed data including multiple data interfaces between the different modelling steps and results is high computational costs. This is especially true if sub-daily simulations are applied that require an additional disaggregation scheme. In contrast, the purely statistical HT-model convinces with its efficient data processing, easily applicable on local computers. A further advantage of the HT-model is the transferability of the approach. While each of the modelling steps of the continuous approach, from weather generator to the hydrological models needs to be implemented, calibrated and validated for every new study area, the HT-model only needs to be fitted to new discharge time series which is less complex. Different advantages and disadvantages of both approaches are finally summarised in Table 1.

**Table 1.** Summary of advantages and disadvantages of the WeGen and HT-model approach to generate heterogeneous flood events.

| Categories | | HT-model | | WeGen |
|---|---|---|---|---|
| Computational complexity | (+) | low processing costs (local processing) | (-) | processing intensive (HPC necessary) |
| | | | (-) | complex data interfaces between different models |
| Output | (-) | Return periods at all sites for modelled events only | (+) | continuous hydrographs at all modelled sites |
| Hydraulic coupling | (-) | event hydrographs need to be deducted to drive a hydraulic model | (+) | continuous description of hydraulic boundary conditions allows unsteady hydraulic modelling |
| Processes | (-) | No information about individual hydrological processes | (+) | continuous description of hydrological system and modelled processes |
| Hydrological changes | (-) | no explicit modelling of scenarios (e.g. climate or land use scenarios) possible | (+) | scenarios can be modelled explicitly (e.g. climate or land use scenarios) |
| | (+) | runoff trends can be integrated | | |
| Transferability | (+) | model is well transferable to other study areas | (-) | model chain is transferable, however all components must be setup and calibrated for new study areas |

Both presented approaches are subject to different uncertainties. The confidence intervals presented in Figure 6 are for example based on the random processes generating heterogeneous flood events of each method (multiple realizations). However, there are other uncertainties which are not explicitly addressed, as for example uncertainties related to the topological kriging of the HT-model results or uncertainties related to the hydrological model in the WeGen approach. Some uncertainties pertain to both methods such as the choice and fitting of the extreme value distributions. A comprehensive assessment by propagating

the uncertainties of all sub-models throughout the model chain is currently precluded by computational constraints particularly relevant for the WeGen approach.

A further important point, currently not considered in both approaches are dike failure scenarios. In the study area, for example no inundation is considered for the River Rhine due to its high protection level. Nonetheless, the probability of a dike failure is non-zero and could have a devastating effect. In this sense, the consideration of flood volumes beside peak estimates could be another important extension to describe the severity of flood events (e.g. Dung et al., 2015; Lamb et al., 2016).

A traditional validation of the overall risk model in terms of a comparison of observed to simulated data is hardly possible as comprehensive databases of loss events are often not available (Thieken et al., 2015). In the present study, damage data based on a insurance portfolio were available for the 2005 event. The data are, however, only a subset of the overall elements at risk and due to rather low sublimits (maximum insurance payout), the full losses remain unknown. Finally, without a larger set of loss events it is not possible to assign a meaningful return period to the 2005 event to validate the risk outcome in a traditional way. Nonetheless, by applying and comparing different methods, the plausibility of the results can be checked (Molinari et al., 2019). Furthermore, the uncertainties related to the choice of methods to generate heterogeneous flood events seem to be lower in comparison to other aspects of the probabilistic flood risk model, such as the choice of the applied damage functions (Winter et al., 2018).

## 6 Conclusions

The question whether the choice of method to generate heterogeneous flood events for flood risk modelling matters can be answered in different ways. Both approaches, the HT-model and continuous WeGen approach, were generally capable of modelling spatially plausible flood events across the study area. By direct comparison to observed spatial patterns, the HT-model approach performed better than the WeGen approach in our study area in terms of correctly representing the observed dependence structure. A stronger modelled dependence of extreme precipitation resulted in high areal rainfall in the WeGen approach and higher overall risk compared to the HT-model. The median damage from 30000 years of simulation is about 17.5% larger in the WeGen approach than in the HT-model. The representation of the dependence structure for simulation of extremes needs to be further improved for the weather generator. Nevertheless, the choice of method to generate heterogeneous flood events might have smaller impact than, for example, the choice of the applied damage functions (Winter et al., 2018).

To conclude, both methods are valid approaches to overcome the simplified assumption of uniform return period across a study area. Accordingly, when designing a flood risk study, the choice of the approach should consider the specific advantages and disadvantages of the two methods and data availability. If computational efficiency and quick transferability are in focus, the HT-model approach might be a better choice. In contrast, if unsteady hydraulic modelling is required for the targeted application, the continuous modelling of generated meteorological fields is more appropriate.

*Code and data availability.* For Austria, daily meteorological and river gauging data are available at https://ehyd.gv.at. The applied meteorological data for the DWD stations are freely available at https://opendata.dwd.de. Underlying loss data are not publicly available. The MeteoIO is available at https://models.slf.ch/p/meteoio/. The applied weather generator and RR-model are currently not publicly available.

*Author contributions.* Based on the initial ideas of KS and SV, the study was designed in collaboration of all authors. BW prepared the initial
5   data, implemented and applied the continuous modelling approach and analysed the results. KF programmed the spatial interpolation scheme for the meteorological data and supported the rainfall-runoff modelling. The risk model and the HT-application were mainly developed by KS. The manuscript was drafted by BW with support of SV. All authors contributed to the review and final version of the manuscript.

*Competing interests.* The authors declare that there is no conflict of interest.

*Acknowledgements.* This work results from the research project HiFlow-CMA [KR15AC8K12522] funded by the Austrian Climate and
10   Energy Fund (ACRP 8th call). We would like to thank all the institutions that provided data, the Zentralanstalt für Meteorologie und Geodynamik (ZAMG), the Deutscher Wetterdienst (DWD), and particularly the Hydrographischer Dienst Vorarlberg. The simulations were conducted using the Vienna Scientific Cluster (VSC). Finally, we want to thank the editors and reviewers for taking their time to critically evaluate this article.

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
