# Peer review of "Event generation for probabilistic flood risk modelling: multi-site peak flow dependence model vs. weather generator based approach"

_Natural Hazards and Earth System Sciences, 2019_

## Short Comment (SC1) · 24 Jan 2020

Dear colleagues,

In your paper, the river discharge approach is referred to as the HT-model approach and the weather generator as the WeGen. As you are aware, we also used the HT model in our river discharge generator (https://www.natural-hazards-and-earth-system-sciences.net/19/1041/2019/). However, our recent weather (precipitation) generator (https://link.springer.com/article/10.1007/s00477-019-01724-9) also makes use of the (purely statistical) HT-model, so this referencing may cause some confusion in the future. It might be better to refer to the river discharge approach as the RDGen approach

or something similar.

Cheers, Dirk Diederen
* * *

---

## Author Comment (AC1) · 29 Jan 2020

Dear Dirk Diederen,

Many thanks for your suggestion. It is a good point that the HT-model can also be applied in context of weather generators, which may cause confusion. We will discuss the wording during the review process.

Best, Benjamin Winter

---

## Referee Comment (RC1) · Anonymous Referee #1 · 5 Feb 2020

Winter et al. presented in their paper ("Event generation for probabilistic flood risk modelling: multi-site peak flow dependence model vs weather generator based approach") two approaches to simulate distributed flood risk throughout rural catchments. The manuscript is well written and structured. The methods are sounds and the models used are well established. The results are clearly presented and the conclusions are supported by the results and discussion. The novelty of the paper is not with the development of new methods, but the use of available methods (that are common in hydrological sciences) in the context of risk assessments. I believe that the application presented here will be of interest to the natural hazard community and fall within the scope of NHESS. Below please find some suggestions for the authors to consider. I

recommend minor revisions.

Specific comments

1. Introduction – You do compare the two distributed approaches to the "traditional" approach, but this is not clear from the introduction. I suggest adding a sentence mentioning this.

2. Discussion – Many other models, besides the HT-model and the WeGen model, can be used to estimate distributed risk. For example, one can use a different WG model (say the AWE-GEN model) and a different hydrological model (say the HBV model) with a different outcome – e.g. that the WG-hydrological model approach will systematically underestimate the risk computed by the HT-model. I suggest adding another paragraph in the discussion section, discussing how general are the results of this study.

3. The WeGen model simulates temperature, but do you use it as input into the hydrological model? It is not clear from the text. If not, I would remove all text mentioning the temperature simulation to avoid confusion.

4. Some justification is needed for the choice of the HQsim model. Is it able to capture well extreme runoff events? Please discuss the advantages and limitations of using a conceptual semi-distributed rainfall-runoff model to simulate floods.

5. [page 7, line 25] Terminology: an ensemble of 100 realizations, each consists of 42 members (years). Also later in the text, replace "repetitions" with "realizations".

6. Figures 3 and 7. Please use a larger font size for the axes labels.

7. [13, 27-31]. I suggest adding in Figure 5 the known losses from the records (e.g. the August 2005 event) and discussing the models' performance in comparison to the "known" risk. It will give another dimension (from an "expert" knowledge) of the abilities of the different models in assessing the risk.

2019-340, 2020.

---

## Referee Comment (RC2) · Martina Kauzlaric (Referee) · 17 Feb 2020

General comments and recommendation

The manuscript by Winter et al. presents an interesting comparison of two quite different approaches for estimating flood risk in a probabilistic framework, both valid and currently established in the research community. The application of these models in a complex environment such as the alpine area and coupling it to a risk assessment is indeed new to my knowledge, and I congratulate the authors for their work! The manuscript is well structured, the methods are described in a comprehensible way or supported by relevant sources, and the discussion provides good points; however,

generally there are quite a lot of relevant numbers /numeric information as well as background information missing. While the authors neatly site sources, they force the reader to go to look for important and relevant information too often, what is laborious and time-consuming first, and an important drawback for both the evaluation and appraisal of the results, resulting in lacking some important considerations in both the results and the discussion parts. This is a pity, because with not too much effort, you might considerably improve the manuscript and better convey relevant take-home messages. The manuscript generally features high-quality and interesting figures, which however would be more easily readable by using larger fonts. In general I found quite a few typing errors. I am reporting all those I found in the technical corrections, but I would generally suggest the authors to read the manuscript thoroughly again. Because of these considerations, I think the manuscript requires further work before it can be recommended for publication. Please find my specific and technical comments here following. Please, don't get scared, some are only suggestions, and some questions are out of curiosity or eventual misunderstanding.

Specific comments

- Introduction: • Please state more clearly the limits and the frames of your application (up to which return period and up to which spatial extent do you think these approaches are applicable and transferable -with this set up? In particular: what do you aim at?) • P2-L17/18: you might want to also cite more recent literature such as Brunner et al. 2019: Modeling the spatial dependence of floods using the Fisher copula, https://doi.org/10.5194/hess-23-107-2019 • P2-L22 Here also there is some more recent literature, such as Evin et al 2018 (Stochastic generation of multi-site daily precipitation focusing on extreme events, https://doi.org/10.5194/hess-22-655-2018) or appeared very recently Raynaud et al.2019 (Assessment of meteorological extremes using a synoptic weather generator and a downscaling model based on analogs, still under discussion, https://doi.org/10.5194/hess-2019-557) • Other two options for generating spatially distributed meteorological fields, more physically based–but also

more computationally intensive-, would be to use the output of either global circula-
tion models (e.g. Felder et al.2018, From global circulation to local flood loss: Cou-
pling models across the scales, https://doi.org/10.1016/j.scitotenv.2018.04.170) or of
hindcast archives (e.g. National Flood Resilience Review. Tech. rep. HM Gov-
ernment, september 2016. url: https://www.gov.uk/government/publications/national-
flood-resilience-review) and downscale these to the required spatial resolution.

- Study area:

• From the map in Figure 1 it seems you are also simulating the Rhine at Lustenau,
is this true? If yes, I assume you are using observations at some gauging station up-
stream of the inflow of the Ill river into the Rhine? If this is the case (and in any case?),
I think it would be quite important to note this later on, as the nodes/communities simu-
lated downstream in the Rhine valley should be considered a bit' differently. - Methods
and Data: • Please introduce how many meteorological stations and gauging sta-
tions are available for this study, for how many years, instead of first mentioning it in
the results part. • Hazard Module II: please give more details about how good is the
modeling chain working (refer also to comments further below under Results). • P6-
L21/22: What do you mean concretely by saying "whereas the underlying hydrological
boundary conditions are based on the considerations of the Austrian flood risk zoning
project HORA"? • Please add some information about the experimental set up: 100
x 42 years for the analysis of the spatial coherence and 30 x 1000 years for the rest
(and also explain why 30 x 1000 years).

- Results: • On the analysis of spatial dependence: Even though it is visible from
the maps you show on the diagonal, it would be more fair to mention (and consider at
all?) in the text that the four stations you are showing in Fig. 3 are not completely "in-
dependent", in the sense that Kennelbach is the downstream station of Thal and in turn
Gisingen is the downstream station of Schruns. If you intentionally chose this set up
–and I could see good reasons for making this choice-, please state it, and explain why.
Furthermore, by reading Winter et al.2019 it occurred to me that first, Thal and Gisingen are actually the two stations with the worst performance both, in the calibration and in the validation periods (if the hydrological model is not able to reproduce well the hydrological features of some subcatchments, depending on the reasons for the low performance, I wouldn't place too much confidence in the results for any other application of this, and rather ask myself if I am ev. not propagating some structural problem in the modeling chain), and second, that apparently both Kennelbach and Gisingen are influenced by hydropower operations (you also state in Winter et al. 2019 "..the influence of hydropower reservoirs cutting peak discharges, especially for the upper Ill catchment. This effect is not considered in the hydrological model set-up, but is contained in the discharge records." => even though I would generally expect this kind of weather generator to overestimate spatial dependence in a complex mountainous environment, this information is relevant when judging the strength of the dependence shown by the WeGen approach). I think you should provide more "background" and or critical information, and accordingly discuss more critically the results. You might be actually attaching too much guilt to the weather generator and/or the WeGen approach. Please do correct me if I am wrong. Another information I am missing here is to what correspond the exceedance probabilities 0.99 and 0.997 (what is the return period we are talking about here? I am not sure I fully understood how you derived the quantiles, sorry if this might be a stupid question). A follow-up consideration: To my knowledge, flood protection measures in Austria are designed whenever possible against a 100 years flood event. For example Felder et al.2017 (The effect of coupling hydrological and hydrodynamic models on maximum flood estimation, http://dx.doi.org/10.1016/j.jhydrol.2017.04.052 ) have shown that there might be considerable potential in re-shaping the hydrograph by coupling a simple 1D hydrodynamic model, in particular in terms of the timing of the peak. While I assume that the effect of retention in the floodplains in your study area is negligible, I would assume that this might become more important downstream for floods with return periods larger than 100 years, let's say for example in the main Rhine valley. As you also look at return periods up to 300 years in the vulnerability module, and you actually make use of inundation maps generated by hydrodynamic models,

when and where do you think the coupling to a hydrodynamic model becomes relevant and what are in this sense the limits of the applicability of the WeGen approach? • Please comment on the larger spread produced by the WeGen approach.

- Discussion: • Weather generators in general: any weather generator makes a quite strong assumption about the tail behaviour, so that the higher the return period resp. the extremeness of the simulated precipitation, the larger should be the structural model uncertainty, which in turn is expected to quite influence the corresponding estimated hydrological load. While a 100 years event might be just at the boundary of what we might be able to extrapolate from about 40 years observations – with still some degree of confidence- anything beyond will very likely be strongly related to the tail models. Could you please elaborate on this, and state what do you think might be the impact of the use of another weather generator on your results? • -P13-L26: Actually in Figure 5 you are showing the "overall" uncertainties of the two modelling chains, what do you mean with and why do you write single uncertainty sources here? • This is just a consideration /suggestion: Of course volumes cannot be considered by applying the HT model, however besides flood peaks, flood volumes can play an important role in flood risk analysis. You correctly mention that one of the advantages of applying WeGen is the ability to produce continuous hydrographs (and accordingly event volumes), however you might want to mention it explicitly? Flood volumes play an important role for hydraulic infrastructure such as reservoirs/lakes/etc.. (and thus in hydraulic design engineering), and also in the case of presence of floodplains with retention potential. On the other side, volumes might be another validation measure for the WeGen approach, as –depending also on how good is working the hydrological model- indirectly indicate how well or bad is the weather generator doing by reproducing persistence at longer time scales (a week and beyond), as I would generally expect this kind of weather generator to be underestimating persistence. This is something you might want to check in the future?

Technical corrections

- Figure 2: it is full of typing errors (refer to Obseravtions, topographie, Geometrie) - Please use the word realizations instead of repetitions - Please use more consistently the word severity (e.g. in the of Figure 4 use return period instead of level of severity => what might be confusing, as you define and quantify severity by the UoFH later on) - P2-L3/4: what do you mean with floods hazard characteristics? - P6-L24: "..a linear interpolated interpolation.." please reformulate better - P7-L8: ..can be statistically.. - P7-L21: dependence matrices instead of dependence metrices - P7-L26: Each simulation instead of Each simulations - P8-L1: the data are too few instead of the data are to few - P10-L6/7: either remove a , in "a significat lower damages.."or change damages to singular - P10: please reformulate the last sentence (90% of exceeding sites sounds weird) - P11-L1: .., simulate (remove a) complex spatially heterogeneous patterns. . . - P11-L5: On the contrary. . .only indirectly .. - P11-L11: just a suggestion=> capability instead of feature? - P11-L13/14: One possible reason could ..be? , - P11-L15/16: might effect ..?=> please reformulate - P12-L2: instead of overall estimation => overestimation ? - P12-L5: ..estimate of (=> better with? Or by?)) WeGen approach - P12-L14: just a suggestion: instead of On the contrary => At the same time? On the other hand?

---

## Author Comment (AC2) · 2 Apr 2020

**Reply to Anonymous Referee #1**

**General Comments**

Winter et al. presented in their paper ("Event generation for probabilistic flood risk modelling: multi-site peak flow dependence model vs weather generator based approach") two approaches to simulate distributed flood risk throughout rural catchments. The manuscript is well written and structured. The methods are sounds and the models used are well established. The results are clearly presented and the conclusions are supported by the results and discussion. The novelty of the paper is not with the development of new methods, but the use of available methods (that are common in hydrological sciences) in the context of risk assessments. I believe that the application presented here will be of interest to the natural hazard community and fall within the scope of NHESS. Below please find some suggestions for the authors to consider. I recommend minor revisions.

> *First of all, we want to thank the anonymous reviewer for taking the time to critically read our work and to provide additional suggestions for further improvement of the manuscript.*

**Specific comments**

1. Introduction – You do compare the two distributed approaches to the "traditional" approach, but this is not clear from the introduction. I suggest adding a sentence mentioning this.

> *Many thanks for this comment. We will add an additional statement in the introduction section to clarify the comparison between the two distributed approaches and the spatially homogeneous approach.*

2. Discussion – Many other models, besides the HT-model and the WeGen model, can be used to estimate distributed risk. For example, one can use a different WG model (say the AWE-GEN model) and a different hydrological model (say the HBV model) with a different outcome – e.g. that the WG-hydrological model approach will systematically underestimate the risk computed by the HT-model. I suggest adding another paragraph in the discussion section, discussing how general are the results of this study.

> *We agree, that the same framework with different components (weather generator or RR-model) likely lead to alternative results. The outcome of higher systematic risk estimates for the WeGen approach might not be true for other model components. We will address this in an additional paragraph in the discussion section.*

3. The WeGen model simulates temperature, but do you use it as input into the hydrological model? It is not clear from the text. If not, I would remove all text mentioning the temperature simulation to avoid confusion.

> *The conceptual RR-Model HQsim is forced by temperature and precipitation. We will add this important information to the "Hazard Module II: WeGen" section.*

4. Some justification is needed for the choice of the HQsim model. Is it able to capture well extreme runoff events? Please discuss the advantages and limitations of using a conceptual semi-distributed rainfall-runoff model to simulate floods.

> *Thank you for the valuable suggestions. The model was used in different studies regarding extreme runoff in alpine study areas and is inter alia applied for the prognosis system of the Inn River (e.g. Senfter et al. 2009; Achleitner et al. 2012; Bellinger et al. 2012; Dobler and Pappenberger 2013; Winter et al. 2019). Fully distributed, physical based models (e.g. WaSim) will probably perform better in describing certain hydrological process such as for example the evapotranspiration or snowmelt process by using energy balance approaches. In contrast, a conceptual description of the hydrological processes (for example in HQsim) does not need all meteorological variables to solve a full energy balance (temperature, precipitation, radiation, humidity and wind speed). A further increase in model complexity will likely compromise the model parameter identifiability, increase calibration effort and computation burden. The*

*computational efficiency is of major concern for the long-term continuous hourly discharge modelling. The advantages and limitations of choosing a conceptual model will be addressed in the discussion section and some further information about HQsim will be added to the method section.*
* * *
5. [page 7, line 25] Terminology: an ensemble of 100 realizations, each consists of 42 members (years). Also later in the text, replace "repetitions" with "realizations".

*We will use the term "realizations" as suggested.*
* * *
6. Figures 3 and 7. Please use a larger font size for the axes labels.

*The Figures will be revised with larger font sizes.*
* * *
7. [13, 27-31]. I suggest adding in Figure 5 the known losses from the records (e.g. the August 2005 event) and discussing the models' performance in comparison to the "known" risk. It will give another dimension (from an "expert" knowledge) of the abilities of the different models in assessing the risk.

*We agree that a "traditional" validation against known losses would be of great value. Unfortunately, we do not have reliable numbers of the loss event which are directly comparable to the model output. We tried to validate the model based on an insurance portfolio. The portfolio is however only a subset of the overall elements at risk and due to rather low sublimits (maximum payouts) for most objects, the full losses remain unknown. Finally, without a larger set of loss events it is not possible to assign a meaningful return period to the 2005 event to "validate" the risk outcome in a traditional way.*

**References**

Achleitner, S*., et al.,* 2012. Analyzing the operational performance of the hydrological models in an alpine flood forecasting system. *Journal of Hydrology,* 412-413, 90–100. doi: 10.1016/j.jhydrol.2011.07.047.

Bellinger, J*., et al.,* 2012. The impact of different elevation steps on simulation of snow covered area and the resulting runoff variance. *Advances in Geosciences,* 32, 69–76. doi: 10.5194/adgeo-32-69-2012.

Dobler, C., and Pappenberger, F., 2013. Global sensitivity analyses for a complex hydrological model applied in an Alpine watershed. *Hydrological Processes,* 27 (26), 3922–3940. doi: 10.1002/hyp.9520.

Senfter, S*., et al.,* 2009. Flood Forecasting for the River Inn. *In:* E. Veulliet, S. Johann, and H. Weck-Hannemann, eds. *Sustainable Natural Hazard Management in Alpine Environments.* Berlin, Heidelberg: Springer Berlin Heidelberg, 35–67. doi: 10.1007/978-3-642-03229-5_2.

Winter, B*., et al.,* 2019. A continuous modelling approach for design flood estimation on sub-daily time scale. *Hydrological Sciences Journal,* 88 (11), 1–16. doi: 10.1080/02626667.2019.1593419.

---

## Author Comment (AC3) · 2 Apr 2020

**Reply to Martina Kauzlaric (Referee #2)**

**General comments and recommendation**

The manuscript by Winter et al. presents an interesting comparison of two quite different approaches for estimating flood risk in a probabilistic framework, both valid and currently established in the research community. The application of these models in a complex environment such as the alpine area and coupling it to a risk assessment is indeed new to my knowledge, and I congratulate the authors for their work! The manuscript is well structured, the methods are described in a comprehensible way or supported by relevant sources, and the discussion provides good points; however, generally there are quite a lot of relevant numbers/numeric information as well as background information missing. While the authors neatly site sources, they force the reader to go to look for important and relevant information too often, what is laborious and time-consuming first, and an important drawback for both the evaluation and appraisal of the results, resulting in lacking some important considerations in both the results and the discussion parts. This is a pity, because with not too much effort, you might considerably improve the manuscript and better convey relevant take-home messages. The manuscript generally features high-quality and interesting figures, which however would be more easily readable by using larger fonts. In general, I found quite a few typing errors. I am reporting all those I found in the technical corrections, but I would generally suggest the authors to read the manuscript thoroughly again. Because of these considerations, I think the manuscript requires further work before it can be recommended for publication. Please find my specific and technical comments here following. Please, don't get scared, some are only suggestions, and some questions are out of curiosity or eventual misunderstanding.

> *First of all, we want to thank Martina Kauzlaric for her comprehensive review with many questions and valuable suggestions to further improve the manuscript. Many thanks also for your positive and motivating words. In this reply, we address all questions and will revise the manuscript accordingly. The Figures will also be revised with larger font sizes to improve the readability.*

**Specific comments**

**Introduction:**

Please state more clearly the limits and the frames of your application (up to which return period and up to which spatial extent do you think these approaches are applicable and transferable -with this set up? In particular: what do you aim at?)

> *Many thanks for this comment. In our opinion there are no definite limits in terms of return period for both approaches. As mentioned in the introduction, risk estimates for large study areas are beside public authorities especially relevant for the (re-)insurance industry. The so called EU Solvency II regulation defines the loss associated with a 0.5% occurrence probability over a one-year period (RP200) as requirement for internal risk management (European Union (EU) 2009). We will add this specific information to the Introduction.*

> *The applicability, however, is strongly depending on available data, whereas in general complexity rises with spatial scale (see Discussion P12-L14/15) and uncertainty rises alongside the extrapolation of the data (this issue will be added in the manuscript). The weather generator approach is currently limited to about 500-600 climate stations and the spatial scale of 1000x1000 km. A high number of climate stations makes the approach computationally intractable. Moreover, the correlation of daily precipitation over distances beyond about 800-1000 km in Central Europe tend to zero on average that results in random spatial precipitation patterns. However, for specific events, the spatial structure still may be retained over large distances.*

P2-L17/18: you might want to also cite more recent literature such as Brunner et al. 2019: Modeling the spatial dependence of floods using the Fisher copula, https://doi.org/10.5194/hess-23-107-2019

> *We will add the citation to the manuscript.*

P2-L22 Here also there is some more recent literature, such as Evin et al 2018 (Stochastic generation of multi-site daily precipitation focusing on extreme events, https://doi.org/10.5194/hess-22-655-2018) or appeared very recently Raynaud et al.2019 (Assessment of meteorological extremes using a synoptic weather generator and a downscaling model based on analogs, still under discussion, https://doi.org/10.5194/hess-2019-557)

*Many thanks for the good literature suggestions. We will add the citations accordingly.*

Other two options for generating spatially distributed meteorological fields, more physically based–but also more computationally intensive-, would be to use the output of either global circulation models (e.g. Felder et al.2018, From global circulation to local flood loss: Coupling models across the scales, https://doi.org/10.1016/j.scitotenv.2018.04.170) or of hindcast archives (e.g. National Flood Resilience Review. Tech. rep. HM Government, September 2016. url: https://www.gov.uk/government /publications/nationalflood- resilience-review) and downscale these to the required spatial resolution.

*Many thanks for this comment. Of course, global and regional climate models represent another way to generate synthetic meteorological fields. However, due to their much longer computational times, only a few realizations of typically about 100 years lengths are feasible. Stochastic weather generators and the HT-model have an advantage to generate hundreds and thousands of possible realizations needed to robustly estimate flood risks. We think, the stochastic methods are still in advantage compared to the climatic models for the purpose of risk assessment.*

**Study area:**

From the map in Figure 1 it seems you are also simulating the Rhine at Lustenau, is this true? If yes, I assume you are using observations at some gauging station upstream of the inflow of the Ill river into the Rhine? If this is the case (and in any case?), I think it would be quite important to note this later on, as the nodes/communities simulated downstream in the Rhine valley should be considered a bit' differently.

*Thank you for this important question. Based on the available risk maps for Vorarlberg, there are no inundation areas designated for the river Rhine due to its high protection level (up to HQ300; see e.g. http://vogis.cnv.at/atlas/init.aspx?karte=wasserbuch&ks=gewaesser, Vorarlberg Map Service in German). The gauge Lustenau (Höchster Brücke) at the river Rhine is actually included in the HT-modelling procedure however is not connected to any community node points and also not included in the hydrological modelling framework of the WeGen approach. Even if they are not considered in this study, there is a risk of dam failures, which could have a devastating effect in Vorarlberg. We will add an additional statement to the Discussion section of the manuscript.*

**Methods and Data:**

Please introduce how many meteorological stations and gauging stations are available for this study, for how many years, instead of first mentioning it in the results part.

*In total 17 gauging stations (1971-2013) are applied for the HT-approach and data of 45 meteorological stations with daily time series from 1971-2013 are included in the WeGen approach. Stations with shorter time series were not considered in the study. We will add this information to the section accordingly.*

Hazard Module II: please give more details about how good is the modeling chain working (refer also to comments further below under Results).

*We will add some additional information regarding the performance of the WeGen, the disaggregation procedure and the hydrological modelling based on the Winter et al. (2019) publication. Accordingly, an additional subchapter will be added to the results section.*

P6-L21/22: What do you mean concretely by saying "whereas the underlying hydrological boundary conditions are based on the considerations of the Austrian flood risk zoning project HORA"?

*As stated, the model chain is not coupled (yet) with a hydraulic model to simulate inundation maps seamlessly. Instead the inundated areas and corresponding water depths for the loss calculation are taken from the "official" inundation maps provided by public authorities. The hydrological loads for the 1D and 2D hydrodynamic simulations are thereby based on the Austrian flood risk zoning project HORA (Merz et al. 2008).*
* * *
Please add some information about the experimental set up: 100 x 42 years for the analysis of the spatial coherence and 30 x 1000 years for the rest (and also explain why 30 x 1000 years).

*For a fair comparison of quantile values between the simulation result and observed data, the time series need to be of identical length. With 42 years of data available at the gauging stations, the simulation was set up accordingly for the analysis of the spatial coherence. As the simulations are theoretically not limited in regard of overall length, all further analyses are based on a total length of 1000 years. The length was chosen to be far above the highest return period of available homogeneous inundation data (HQ300) and the number of 30 realizations were calculated due to the computational limitations of the continuous simulation on an hourly time step.*

**Results:**

On the analysis of spatial dependence: Even though it is visible from the maps you show on the diagonal, it would be more fair to mention (and consider at all?) in the text that the four stations you are showing in Fig. 3 are not completely "independent", in the sense that Kennelbach is the downstream station of Thal and in turn Gisingen is the downstream station of Schruns. If you intentionally chose this set up –and I could see good reasons for making this choice-, please state it, and explain why.

*Thank you for this comment. We choose the gauges Kennelbach and Gisingen as they are the gauges closest to the outlets of the two largest catchments in the study area. This two catchments comprising about 80% of the total study area. We intentionally choose Schruns and Thal which are subcatchments of Gisingen and Kennelbach, respectively. By choosing this examples we show two strongly related gauges in nested catchments (but never completely dependent) as well as two relatively independent gauging stations (e.g. Schruns and Thal). We will add an additional explanation for the choice to the manuscript.*

Furthermore, by reading Winter et al. 2019 it occurred to me that first, Thal and Gisingen are actually the two stations with the worst performance both, in the calibration and in the validation periods (if the hydrological model is not able to reproduce well the hydrological features of some subcatchments, depending on the reasons for the low performance, I wouldn't place too much confidence in the results for any other application of this, and rather ask myself if I am ev. not propagating some structural problem in the modeling chain), and second, that apparently both Kennelbach and Gisingen are influenced by hydropower operations (you also state in Winter et al. 2019 "...the influence of hydropower reservoirs cutting peak discharges, especially for the upper Ill catchment. This effect is not considered in the hydrological model set-up, but is contained in the discharge records." => even though I would generally expect this kind of weather generator to overestimate spatial dependence in a complex mountainous environment, this information is relevant when judging the strength of the dependence shown by the WeGen approach). I think you should provide more "background" and or critical information, and accordingly discuss more critically the results. You might be actually attaching too much guilt to the weather generator and/or the WeGen approach.

*It is correct that the models at Thal and Gisingen do not perform well. Both the spatial dependence and the hydrological model deficiency play a role in poor performance at the Thal and Gisingen. However, the tendency to overestimate the spatial dependence in comparison to the observed data is present for many station pairs as for example the gauges "Hoher Steg" (NSE 0.85/0.80) and "Kennelbach" (NSE 0.73/0.69) shown in Figure 1. We agree that some further information regarding the performance of the modelling chain will improve the manuscript and as mentioned above we will add some additional information based on the Winter et al. (2019).*

[Figure]

*Figure 1 Spatial dependence measure observed vs. WeGen approach; conditioning site Kennelbach, dependent site Hoher Steg.*

Please do correct me if I am wrong. Another information I am missing here is to what correspond the exceedance probabilities 0.99 and 0.997 (what is the return period we are talking about here? I am not sure I fully understood how you derived the quantiles, sorry if this might be a stupid question).

> *This is actually a very good question. The thresholds are based on the quantile values of the time series. So, in the example shown in figure 3, the 0.99 to 0.997 quantile values are based on 42 years of input data. As the data analysis is based on 3-day block maxima values to guarantee the independence of events (Schneeberger and Steinberger 2018). Accordingly, based on the empirical cdf, a p-value of 0.99 refers to a return period of approximately 1 year and a p-value of 0.997 refers to a return period of roughly 3 years. We will revise the text accordingly, for a better understanding of the quantile values.*

A follow-up consideration: To my knowledge, flood protection measures in Austria are designed whenever possible against a 100 years flood event. For example, Felder et al.2017 (The effect of coupling hydrological and hydrodynamic models on maximum flood estimation, http://dx.doi.org/10.1016/j.jhydrol.2017.04.052) have shown that there might be considerable potential in re-shaping the hydrograph by coupling a simple 1D hydrodynamic model, in particular in terms of the timing of the peak. While I assume that the effect of retention in the floodplains in your study area is negligible, I would assume that this might become more important downstream for floods with return periods larger than 100 years, let's say for example in the main Rhine valley. As you also look at return periods up to 300 years in the vulnerability module, and you actually make use of inundation maps generated by hydrodynamic models, when and where do you think the coupling to a hydrodynamic model becomes relevant and what are in this sense the limits of the applicability of the WeGen approach?

> *Yes, we do agree with this comment that the so-called hydrodynamic interactions in the river network may affect the risk estimates, i.e. dike overtopping and failure upstream with associated inundation and water storage would reduce the risk downstream. The higher the return period of evet is, the stronger the effect of hydrodynamic interactions is expected to be. Similarly, the larger the potential storage area in the hinterland is, which is the case for the lowland parts of the river network, the stronger the effect is. By including a 1D-2D hydrodynamic modelling, the hydrodynamic interactions can be explicitly considered in the WeGen approach. This could be a potential future extension of the modelling approach, particularly for the lowland parts of the network. On the contrary, the HT-approach is not suitable for coupling with the unsteady continuous hydrodynamic models, since it is not mass-conservative and delivers only dependent discharge peaks and not the full continuous flood hydrographs as boundary conditions. We will provide this discussion in the revised manuscript.*

Please comment on the larger spread produced by the WeGen approach.

*The weather generator tends to overestimate the spatial correlation of extreme precipitation. It is parameterized by using an isotropic correlation function by mixing low intensity large scale and high intensity local rainfalls (see P11.L16 - P12.L5). Hence, the generated fields of extreme precipitation tend to have larger spatial extent than observed.*

**Discussion:**

Weather generators in general: any weather generator makes a quite strong assumption about the tail behaviour, so that the higher the return period resp. the extremeness of the simulated precipitation, the larger should be the structural model uncertainty, which in turn is expected to quite influence the corresponding estimated hydrological load. While a 100 years event might be just at the boundary of what we might be able to extrapolate from about 40 years observations – with still some degree of confidence- anything beyond will very likely be strongly related to the tail models. Could you please elaborate on this, and state what do you think might be the impact of the use of another weather generator on your results?

*We agree that to do the extrapolation weather generators make strong assumptions about the tail distribution and such the uncertainty raises alongside the exceedance probability which is directly propagated to the hydrological loads. The extrapolation beyond RPs of 100 years based on typically available data series of a few decades is associated with large uncertainties. Nonetheless, information about higher return periods are often required in practice (e.g. In Austria HQ300 is applied to define "residual risk areas" or the RP of 200 years is defined in the Solvency II definition in the European (re)insurance context (European Union (EU) 2009)). To our knowledge, there are no studies comparing risk assessments driven by different weather generators. Hence, it is difficult to make a reliable statement how decisive the tail dependence is with regards to the final risk estimates. On the one side, the effect of tail dependence is expected to increase with the return period. On the other side, the events with high return periods have low probability and might have little impact on the average risk (expected annual damage) (area under the risk curve). So, this is a question whether we look at the loss estimate of a e.g. 1000-year flood or we are interested in the annual expected damage. For the first, the tail dependence might be more important, for the second rather less important. We believe, more studies are needed to compare different weather generators and their impact on risk assessments. We will address this issue in an additional paragraph in the discussion section.*

P13-L26: Actually in Figure 5 you are showing the "overall" uncertainties of the two modelling chains, what do you mean with and why do you write single uncertainty sources here?

*As stated on P.13-L20-21, the uncertainty presented in Figure 5 only shows the uncertainty which corresponds to the multiple realizations and does not account for other sources of uncertainties (e.g. parameter uncertainties). Also, no comprehensive uncertainty assessment by propagating the uncertainties of all sub-models throughout the model chain is included in the current work, it is still possible to have a look at each individual modelling step. We will reformulate the statement accordingly.*

This is just a consideration /suggestion: Of course volumes cannot be considered by applying the HT model, however besides flood peaks, flood volumes can play an important role in flood risk analysis. You correctly mention that one of the advantages of applying WeGen is the ability to produce continuous hydrographs (and accordingly event volumes), however you might want to mention it explicitly? Flood volumes play an important role for hydraulic infrastructure such as reservoirs/lakes/etc. (and thus in hydraulic design engineering), and also in the case of presence of floodplains with retention potential. On the other side, volumes might be another validation measure for the WeGen approach, as –depending also on how good is working the hydrological model- indirectly indicate how well or bad is the weather generator doing by reproducing persistence at longer time scales (a week and beyond), as I would generally expect this kind of weather generator to be underestimating persistence. This is something you might want to check in the future?

*Many thanks for this interesting suggestion. We agree that flood volumes are an important characteristic of flood events and especially relevant for risk assessment (e.g. Dung et al. 2015; Lamb et al. 2016). We will elaborate more on this in the discussion. Future research can focus on the ability of the model chain WeGen-RR to reproduce flood volumes. This is however not straightforward. This can be approached by the assessment of the areal precipitation volume for different durations and spatial aggregations (from small sub-catchments to the entire basin)*

*and secondly, by comparing the observed and simulated flood event volume statistics similarly to the flood peak statistics. We will add this notion in the discussion chapter.*

**Technical corrections**

Figure 2: it is full of typing errors (refer to Obseravtions, topographie, Geometrie)

*The Figure will be revised accordingly.*
* * *
Please use the word realizations instead of repetitions

*We will use the term "realizations" throughout the manuscript.*
* * *
Please use more consistently the word severity (e.g. in the of Figure 4 use return period instead of level of severity => what might be confusing, as you define and quantify severity by the UoFH later on)

*As suggested the caption was changed to return period to avoid confusion.*
* * *
P2-L3/4: what do you mean with floods hazard characteristics?

*For example, inundation depth or flow velocity. This information will be added for clarification.*
* * *
P6-L24: "...a linear interpolated interpolation..." please reformulate better

*The term "interpolated" will be deleted.*
* * *
P7-L8: ...can be statistically.

*"statistically" will be added as suggested.*

P10: please reformulate the last sentence (90% of exceeding sites sounds weird)

*The sentence will be revised to: "The most widespread event corresponds to about 90% of the sites exceeding the threshold (UoFH=77)."*
* * *
P11-L11: just a suggestion=> capability instead of feature?

*Thank you for the suggestion. We will use the term "capability" instead of feature.*
* * *
P11-L15/16: might effect ...?=> please reformulate

*'effect' will be replaced by 'affect'*
* * *
P12-L5: …estimate of (=> better with? Or by?)) WeGen approach

*The manuscript will be changed to "...estimate by the WeGen approach."*
* * *
P12-L14: just a suggestion: instead of On the contrary => At the same time? On the other hand?

*Thank you for the suggestion. The phrase "At the same time" will be used instead.*
* * *
P7-L21: dependence matrices instead of dependence metrices

P7-L26: Each simulation instead of Each simulations

P8-L1: the data are "too" few instead of the data are "to" few

P10-L6/7: either remove a , in "a significat lower damages.."or change damages to singular

P11-L1: …simulate (remove a) complex spatially heterogeneous patterns.

P11-L5: On the contrary. . .only indirectly …

P11-L13/14: One possible reason could ...be?

P12-L2: instead of overall estimation => overestimation?

> *Thank you for the grammar and language corrections. We will revise the manuscript accordingly.*

**References**

Dung, N.V*., et al.,* 2015. Handling uncertainty in bivariate quantile estimation – An application to flood hazard analysis in the Mekong Delta. *Journal of Hydrology,* 527, 704–717. doi: 10.1016/j.jhydrol.2015.05.033.

European Union (EU), 2009. on the taking-up and pursuit of the business of Insurance and Reinsurance (Solvency II): Directive 2009/138/EC of the European Parliament and the Council. *Official Journal of the European Union (*L335).

Lamb, R*., et al.,* 2016. Have applications of continuous rainfall-runoff simulation realized the vision for process-based flood frequency analysis? *Hydrological Processes,* 30 (14), 2463–2481. doi: 10.1002/hyp.10882.

Merz, R., Blöschl, G., and Humer, G., 2008. National flood discharge mapping in Austria. *Natural Hazards,* 46 (1), 53–72. doi: 10.1007/s11069-007-9181-7.

Schneeberger, K*., et al.,* 2019. A Probabilistic Framework for Risk Analysis of Widespread Flood Events. A Proof-of-Concept Study. *Risk Analysis,* 39 (1), 125-139. doi: 10.1111/risa.12863.

Schneeberger, K., and Steinberger, T., 2018. Generation of Spatially Heterogeneous Flood Events in an Alpine Region—Adaptation and Application of a Multivariate Modelling Procedure. *Hydrology,* 5 (1), 5. doi: 10.3390/hydrology5010005.

Winter, B*., et al.,* 2019. A continuous modelling approach for design flood estimation on sub-daily time scale. *Hydrological Sciences Journal,* 88 (11), 1–16. doi: 10.1080/02626667.2019.1593419.

---

## Author Response (AR1)

manuscript nhess-2019-340

**Cover Letter and Reply to the Referees**

Dear Prof. Keiler,

we would like to thank you and the two reviewers (Martina Kauzlaric and anonymous) for taking the time to critically read our manuscript and to provide valuable and constructive feedback. We have addressed the comments and suggestions of referees in this document. Among others, we provide an additional subchapter about the performance of the continuous modelling approach (WeGen), extended the discussion in various points following the suggestions of the reviewers and revised the figures with larger font sizes to improve the readability. We hope the revised manuscript has further improved by the comments and suggestions.

Please, find below the reply to the reviewers and your additional questions (answers in italics). In addition, a separate track change manuscript file is attached to this document.

As discussed, we decided to add Kristian Förster as a co-author to the manuscript. Although his contribution was important, in particular in the early stage of our research, we unfortunately did not include him as a co-author. This was my personal mistake as leading author.

Kristian contributed to the development of the continuous modelling chain and the rainfall runoff modelling. He was responsible for the development and programming of the spatial interpolation scheme of the meteorological data. Finally, he took part in the revision process. To improve transparency, the contribution section of the manuscript was revised with more details of each individual role.

Yours sincerely,
Benjamin Winter

Also on behalf of my co-authors Klaus Schneeberger, Kristian Förster, and Sergiy Vorogushyn.
* * *
**Revised Authors contribution:**

*Based on the initial ideas of KS and SV, the study was designed in collaboration of all authors. BW prepared the initial data, implemented and applied the continuous modelling approach and analysed the results. KF programmed the spatial interpolation scheme for the meteorological data and supported the rainfall-runoff modelling. The risk model and the HT-application were mainly developed by KS. The manuscript was drafted by BW with support of SV. All authors contributed to the review and final version of the manuscript.*

**Reply to Margreth Keiler, Editor**

a) Damage has no plural and thus the term 'damages' has a different meaning. please check and adapt accordingly

*The manuscript was changed accordingly.*

b) The second sentences in your abstracts is not clear if you consider the risk definition your study is based on.

*Thank you for this comment. We reformulated the second sentence in the abstract to: "To estimate the risk, i.e. the probability of damage, flood damage needs to be either systematically recorded over long period or it needs to be modelled for a series of synthetically generated flood events" In addition, we extended the paragraph in the introduction.*

c) Please provide more information on section 3.3 and 3.4 because without reading Schneeberger et al. (2019) this part is not clear. Furthermore, why did you chose the approach of Borter (1999) which overestimates the risk, and did not consider models like FLEMOps or others?

*Some further information was added to section 3.3 and 3.4. The damage model of Borter (1999) was chosen as it originates from Switzerland which is a direct neighbor to the Austrian province Vorarlberg with a similar topographic situation and building structure. Nonetheless, other damage models such as FLEMO could have been used as well. The study rather focuses on the comparison of the results between the two different approaches for the generation of flood event series. Nonetheless, in the best case local functions could be derived, however, the data basis is not available. The notion about the choice of the damage model was also added to the manuscript.*

d) Please provide more information why you think a comparison on risk level allows more insights than on hazard level. In this context, I miss a discussion on the effect of the spatial distributions of elements at risk to your results.

*Thank you for this good comment. In the study area, the settlements are concentrated alongside the larger valley areas (especially the Rhine valley; c.f. map Figure 1). Thus, the damage corresponding to an event is largely influenced by the region affected. If the comparison is conducted on a hazard only, the impact of wide spread flood events may be overestimated, while the impact of spatially limited events in densely populated areas are underestimated. This discussion was added to the Manuscript (P.13 L5-9)*

e) For any currency values, both in text, tables, and figures, tell the reader what year these have been normalized to.

*Thank you for this comment. The building values are indexed to the year 2013 (Huttenlau et al. 2015) based on the austrian construction price index (Statistik Austria 2019). This information was added to the section 3.3 and to the caption Figure 6 in the manuscript.*

f) Page 10, line 4: please adapt the equation format according to the guidelines

*The format of the equation was adapted according to the guidelines on "simple expressions in the body of the text"*

**Interactive comment, Dirk Diederen**

In your paper, the river discharge approach is referred to as the HT-model approachand the weather generator as the WeGen. As you are aware, we also used the HTmodel in our river discharge generator (https://www.natural-hazards-and-earth-system-sciences.net/19/1041/2019/). However, our recent weather (precipitation) generator(https://link.springer.com/article/10.1007/s00477-019-01724-9) also makes use of the (purely statistical) HT-model, so this referencing may cause some confusion in the future. It might be better to refer to the river discharge approach as the RDGen approachC1or something similar.

> *We thank Dirk Diederen for his notion. We can in principle follow his argument as the HT model is a general statistical model to describe tail dependence. It can be applied to different variables besides discharge peaks. In your cited paper (Diederen et al., 2019), the HT model is however one of the pieces of a larger model to generate synthetic events, which includes e.g. spatial event identification, derivation of event characteristics and construction of new spatially consistent events. In their case, it would indeed be not suitable to name the entire model as HT model. In our case, HT model is used to describe dependence of peak discharges and construct synthetic scenarios and basically represents the hazard module. We indicate that the HT model is used as a hazard module within the PRAMO – probabilistic risk model. WeGen is an alternative hazard module based on a weather generator approach. We therefore would like to keep this notion of 'HTm' referring solely to the hazard module, also to be consistent with the previous works of Schneeberger et al. (2018, 2019), who described in details the setup of the HT model as a part of the more comprehensive PRAMo modelling framework using the very same notation.*

**Reply to Anonymous Referee #1**

**General Comments**

Winter et al. presented in their paper ("Event generation for probabilistic flood risk modelling: multi-site peak flow dependence model vs weather generator based approach") two approaches to simulate distributed flood risk throughout rural catchments. The manuscript is well written and structured. The methods are sounds and the models used are well established. The results are clearly presented and the conclusions are supported by the results and discussion. The novelty of the paper is not with the development of new methods, but the use of available methods (that are common in hydrological sciences) in the context of risk assessments. I believe that the application presented here will be of interest to the natural hazard community and fall within the scope of NHESS. Below please find some suggestions for the authors to consider. I recommend minor revisions.

> *First of all, we want to thank the anonymous reviewer for taking the time to critically read our work and to provide additional suggestions for further improvement of the manuscript. We addressed the comments in the revised manuscript.*

**Specific comments**

1. Introduction – You do compare the two distributed approaches to the "traditional" approach, but this is not clear from the introduction. I suggest adding a sentence mentioning this.

> *Many thanks for this comment. We revised the sentence in the introduction section to clarify flood loss is also compared to a homogeneous flood scenario is named "traditional" approach.*
>
> *Please see P3. L.3-4: "Additionally, the flood risk corresponding to homogeneous flood scenarios of certain return periods ("traditional" approach) is derived and compared to the other two approaches."*

2. Discussion – Many other models, besides the HT-model and the WeGen model, can be used to estimate distributed risk. For example, one can use a different WG model (say the AWE-GEN model) and a different hydrological model (say the HBV model) with a different outcome – e.g. that the WG-hydrological model approach will systematically underestimate the risk computed by the HT-model. I suggest adding another paragraph in the discussion section, discussing how general are the results of this study.

> *We agree, that the same framework with different components (weather generator or RR-model) likely lead to alternative results. The outcome of higher systematic risk estimates for the WeGen approach might not be true for other model components. We addressed this important issue in an additional paragraph in the discussion section.*

3. The WeGen model simulates temperature, but do you use it as input into the hydrological model? It is not clear from the text. If not, I would remove all text mentioning the temperature simulation to avoid confusion.

> *The conceptual RR-Model HQsim is forced by temperature and precipitation. The information was added to the "Hazard Module II: WeGen" section.*

4. Some justification is needed for the choice of the HQsim model. Is it able to capture well extreme runoff events? Please discuss the advantages and limitations of using a conceptual semi-distributed rainfall-runoff model to simulate floods.

> *Thank you for the valuable suggestions. The model was used in different studies regarding extreme runoff in alpine study areas and is inter alia applied for the prognosis system of the Inn River (e.g. Senfter et al. 2009; Achleitner et al. 2012; Bellinger et al. 2012; Dobler and Pappenberger 2013;*

*Winter et al. 2019). An additional subchapter was added to summarizing information regarding the performance of the weather generator, the disaggregation procedure and the hydrological modelling (see also Referee#2).*

*Fully distributed, physical based models (e.g. WaSim) will probably perform better in describing certain hydrological process such as for example the evapotranspiration or snowmelt process by using energy balance approaches. In contrast, a conceptual description of the hydrological processes (for example in HQsim) does not need all meteorological variables to solve a full energy balance (temperature, precipitation, radiation, humidity and wind speed). A further increase in model complexity will likely compromise the model parameter identifiability, increase calibration effort and computation burden. The computational efficiency is of major concern for the long-term continuous hourly discharge modelling. The advantages and limitations of choosing a conceptual model are now addressed in the discussion section. (see P.15)*
* * *
5. [page 7, line 25] Terminology: an ensemble of 100 realizations, each consists of 42 members (years). Also later in the text, replace "repetitions" with "realizations".

*The term "realizations" was used as suggested.*
* * *
6. Figures 3 and 7. Please use a larger font size for the axes labels.

*All Figures have been revised with larger font sizes to increase readability.*
* * *
7. [13, 27-31]. I suggest adding in Figure 5 the known losses from the records (e.g. the August 2005 event) and discussing the models' performance in comparison to the "known" risk. It will give another dimension (from an "expert" knowledge) of the abilities of the different models in assessing the risk.

*We agree that a "traditional" validation against known losses would be of great value. Unfortunately, we do not have reliable numbers of the loss event which are directly comparable to the model output. We tried to validate the model based on an insurance portfolio. The portfolio is however only a subset of the overall elements at risk and due to rather low sublimits (maximum payouts) for most objects, the full losses remain unknown. Finally, without a larger set of loss events it is not possible to assign a meaningful return period to the 2005 event to "validate" the risk outcome in a traditional way. This consideration was added to the "Discussion" section (P.17 L8-12).*

**Reply to Martina Kauzlaric (Referee #2)**

**General comments and recommendation**

The manuscript by Winter et al. presents an interesting comparison of two quite different approaches for estimating flood risk in a probabilistic framework, both valid and currently established in the research community. The application of these models in a complex environment such as the alpine area and coupling it to a risk assessment is indeed new to my knowledge, and I congratulate the authors for their work! The manuscript is well structured, the methods are described in a comprehensible way or supported by relevant sources, and the discussion provides good points; however, generally there are quite a lot of relevant numbers/numeric information as well as background information missing. While the authors neatly site sources, they force the reader to go to look for important and relevant information too often, what is laborious and time-consuming first, and an important drawback for both the evaluation and appraisal of the results, resulting in lacking some important considerations in both the results and the discussion parts. This is a pity, because with not too much effort, you might considerably improve the manuscript and better convey relevant take-home messages. The manuscript generally features high-quality and interesting figures, which however would be more easily readable by using larger fonts. In general, I found quite a few typing errors. I am reporting all those I found in the technical corrections, but I would generally suggest the authors to read the manuscript thoroughly again. Because of these considerations, I think the manuscript requires further work before it can be recommended for publication. Please find my specific and technical comments here following. Please, don't get scared, some are only suggestions, and some questions are out of curiosity or eventual misunderstanding.

> *First of all, we want to thank Martina Kauzlaric for her comprehensive review with many questions and valuable suggestions to further improve the manuscript. Many thanks also for your positive and motivating words. In this reply, we address all questions and revised the manuscript accordingly. The Figures have also been revised with larger font sizes to improve the readability.*

**Specific comments**

**Introduction:**

Please state more clearly the limits and the frames of your application (up to which return period and up to which spatial extent do you think these approaches are applicable and transferable -with this set up? In particular: what do you aim at?)

> *Many thanks for this comment. In our opinion there are no definite limits in terms of return period for both approaches. As mentioned in the introduction, risk estimates for large study areas are beside public authorities especially relevant for the (re-)insurance industry. The so called EU Solvency II regulation defines the loss associated with a 0.5% occurrence probability over a one-year period (RP200) as requirement for internal risk management (European Union (EU) 2009). This specific information was added to the Introduction.*

> *The applicability, however, is strongly depending on available data, whereas in general complexity rises with spatial scale (see Discussion P12-L14/15) and uncertainty rises alongside the extrapolation of the data (this issue was added to the subchapter "Comparison of risk curves"). The weather generator approach is currently limited to about 500-600 climate stations and the spatial scale of 1000x1000 km. A high number of climate stations makes the approach computationally intractable. Moreover, the correlation of daily precipitation over distances beyond about 800-1000 km in Central Europe tend to zero on average that results in random spatial precipitation patterns. However, for specific events, the spatial structure still may be retained over large distances.*

P2-L17/18: you might want to also cite more recent literature such as Brunner et al. 2019: Modeling the spatial dependence of floods using the Fisher copula, https://doi.org/10.5194/hess-23-107-2019

> *We added the citation to the manuscript.*

P2-L22 Here also there is some more recent literature, such as Evin et al 2018 (Stochastic generation of multi-site daily precipitation focusing on extreme events, https://doi.org/10.5194/hess-22-655-2018) or appeared very recently Raynaud et al.2019 (Assessment of meteorological extremes using a synoptic weather generator and a downscaling model based on analogs, still under discussion, https://doi.org/10.5194/hess-2019-557)

*Many thanks for the good literature suggestions. The citations were added accordingly.*
* * *
Other two options for generating spatially distributed meteorological fields, more physically based–but also more computationally intensive-, would be to use the output of either global circulation models (e.g. Felder et al.2018, From global circulation to local flood loss: Coupling models across the scales, https://doi.org/10.1016/j.scitotenv.2018.04.170) or of hindcast archives (e.g. National Flood Resilience Review. Tech. rep. HM Government, September 2016. url: https://www.gov.uk/government /publications/nationalflood-resilience-review) and downscale these to the required spatial resolution.

*Many thanks for this comment. Of course, global and regional climate models represent another way to generate synthetic meteorological fields. However, due to their much longer computational times, only a few realizations of typically about 100 years lengths are feasible. Stochastic weather generators and the HT-model have an advantage to generate hundreds and thousands of possible realizations needed to robustly estimate flood risks. We think, the stochastic methods are still in advantage compared to the climatic models for the purpose of risk assessment.*

**Study area:**

From the map in Figure 1 it seems you are also simulating the Rhine at Lustenau, is this true? If yes, I assume you are using observations at some gauging station upstream of the inflow of the Ill river into the Rhine? If this is the case (and in any case?), I think it would be quite important to note this later on, as the nodes/communities simulated downstream in the Rhine valley should be considered a bit' differently.

*Thank you for this important question. Based on the available risk maps for Vorarlberg, there are no inundation areas designated for the river Rhine due to its high protection level (up to HQ300; see e.g. http://vogis.cnv.at/atlas/init.aspx?karte=wasserbuch&ks=gewaesser, Vorarlberg Map Service in German). The gauge Lustenau (Höchster Brücke) at the river Rhine is actually included in the HT-modelling procedure however is not connected to any community node points and also not included in the hydrological modelling framework of the WeGen approach. Even if they are not considered in this study, there is a risk of dam failures, which could have a devastating effect in Vorarlberg. We added an additional statement to the Discussion section of the manuscript (P.17 L3-7).*

**Methods and Data:**

Please introduce how many meteorological stations and gauging stations are available for this study, for how many years, instead of first mentioning it in the results part.

*In total 17 gauging stations (1971-2013) are applied for the HT-approach and data of 45 meteorological stations with daily time series from 1971-2013 are included in the WeGen approach. Stations with shorter time series were not considered in the study. This information was added to the Methods and Data section accordingly. (P.4 L1-4)*
* * *
Hazard Module II: please give more details about how good is the modeling chain working (refer also to comments further below under Results).

*An additional subchapter was added summarizing the results of the Hazard Module II (WeGen). In the subchapter information regarding the performance of the weather generator, the disaggregation procedure and the hydrological modelling are included. More detailed information is given in Winter et al. (2019).*

P6-L21/22: What do you mean concretely by saying "whereas the underlying hydrological boundary conditions are based on the considerations of the Austrian flood risk zoning project HORA"?

> *As stated, the model chain is not coupled (yet) with a hydraulic model to simulate inundation maps seamlessly. Instead the inundated areas and corresponding water depths for the loss calculation are taken from the "official" inundation maps provided by public authorities. The hydrological loads for the 1D and 2D hydrodynamic simulations are thereby based on the Austrian flood risk zoning project HORA (Merz et al. 2008). The paragraph was rephrased accordingly.*
* * *
Please add some information about the experimental set up: 100 x 42 years for the analysis of the spatial coherence and 30 x 1000 years for the rest (and also explain why 30 x 1000 years).

> *For a fair comparison of quantile values between the simulation result and observed data, the time series need to be of identical length. With 42 years of data available at the gauging stations, the simulation was set up accordingly for the analysis of the spatial coherence (See P.9 L6-7).*

> *As the simulations are theoretically not limited in regard of overall length, all further analyses are based on a total length of 1000 years. The length was chosen to be far above the highest return period of available homogeneous inundation data (HQ300) and the number of 30 realizations were calculated due to the computational limitations of the continuous simulation on an hourly time step. This information was added to the corresponding result section (See P.11 L 7-11).*

**Results:**

On the analysis of spatial dependence: Even though it is visible from the maps you show on the diagonal, it would be more fair to mention (and consider at all?) in the text that the four stations you are showing in Fig. 3 are not completely "independent", in the sense that Kennelbach is the downstream station of Thal and in turn Gisingen is the downstream station of Schruns. If you intentionally chose this set up –and I could see good reasons for making this choice-, please state it, and explain why.

> *Thank you for this comment. We choose the gauges Kennelbach and Gisingen as they are the gauges closest to the outlets of the two largest catchments in the study area. This two catchments comprising about 80% of the total study area. We intentionally choose Schruns and Thal which are subcatchments of Gisingen and Kennelbach, respectively. By choosing this examples we show two strongly related gauges in nested catchments (but never completely dependent) as well as two relatively independent gauging stations (e.g. Schruns and Thal). The explanation for the choice of the gauges has been added to the manuscript. (P.9 L8- P.10 L2)*

Furthermore, by reading Winter et al. 2019 it occurred to me that first, Thal and Gisingen are actually the two stations with the worst performance both, in the calibration and in the validation periods (if the hydrological model is not able to reproduce well the hydrological features of some subcatchments, depending on the reasons for the low performance, I wouldn't place too much confidence in the results for any other application of this, and rather ask myself if I am ev. not propagating some structural problem in the modeling chain), and second, that apparently both Kennelbach and Gisingen are influenced by hydropower operations (you also state in Winter et al. 2019 "...the influence of hydropower reservoirs cutting peak discharges, especially for the upper Ill catchment. This effect is not considered in the hydrological model set-up, but is contained in the discharge records." => even though I would generally expect this kind of weather generator to overestimate spatial dependence in a complex mountainous environment, this information is relevant when judging the strength of the dependence shown by the WeGen approach). I think you should provide more "background" and or critical information, and accordingly discuss more critically the results. You might be actually attaching too much guilt to the weather generator and/or the WeGen approach.

> *It is correct that the models at Thal and Gisingen do not perform well. Both the spatial dependence and the hydrological model deficiency play a role in poor performance at the Thal and Gisingen. However, the tendency to overestimate the spatial dependence in comparison to the observed data*

*is present for many station pairs as for example the gauges "Hoher Steg" (NSE 0.85/0.80) and "Kennelbach" (NSE 0.73/0.69) shown in Figure 1.*

*We agree that further information regarding the performance of the modelling chain will improve the manuscript. An additional subchapter was added to the "results" containing information about the performance of the weather generator, disaggregation procedure and the rainfall runoff model based on the Winter et al. (2019).*

[Figure]

*Figure 1 Spatial dependence measure observed vs. WeGen approach; conditioning site Kennelbach, dependent site Hoher Steg.*

Please do correct me if I am wrong. Another information I am missing here is to what correspond the exceedance probabilities 0.99 and 0.997 (what is the return period we are talking about here? I am not sure I fully understood how you derived the quantiles, sorry if this might be a stupid question).

*This is actually a very good question. The thresholds are based on the quantile values of the time series. So, in the example shown in figure 3, the 0.99 to 0.997 quantile values are based on 42 years of input data. As the data analysis is based on 3-day block maxima values to guarantee the independence of events (Schneeberger and Steinberger 2018). Accordingly, based on the empirical cdf, a p-value of 0.99 refers to a return period of approximately 1 year and a p-value of 0.997 refers to a return period of roughly 3 years. The text was revised accordingly, for a better understanding of the quantile values. (P.10)*

A follow-up consideration: To my knowledge, flood protection measures in Austria are designed whenever possible against a 100 years flood event. For example, Felder et al.2017 (The effect of coupling hydrological and hydrodynamic models on maximum flood estimation, http://dx.doi.org/10.1016/j.jhydrol.2017.04.052) have shown that there might be considerable potential in re-shaping the hydrograph by coupling a simple 1D hydrodynamic model, in particular in terms of the timing of the peak. While I assume that the effect of retention in the floodplains in your study area is negligible, I would assume that this might become more important downstream for floods with return periods larger than 100 years, let's say for example in the main Rhine valley. As you also look at return periods up to 300 years in the vulnerability module, and you actually make use of inundation maps generated by hydrodynamic models, when and where do you think the coupling to a hydrodynamic model becomes relevant and what are in this sense the limits of the applicability of the WeGen approach?

*Yes, we do agree with this comment that the so-called hydrodynamic interactions in the river network may affect the risk estimates, i.e. dike overtopping and failure upstream with associated inundation and water storage would reduce the risk downstream. The higher the return period of evet is, the*

*stronger the effect of hydrodynamic interactions is expected to be. Similarly, the larger the potential storage area in the hinterland is, which is the case for the lowland parts of the river network, the stronger the effect is. By including a 1D-2D hydrodynamic modelling, the hydrodynamic interactions can be explicitly considered in the WeGen approach. This could be a potential future extension of the modelling approach, particularly for the lowland parts of the network. On the contrary, the HT-approach is not suitable for coupling with the unsteady continuous hydrodynamic models, since it is not mass-conservative and delivers only dependent discharge peaks and not the full continuous flood hydrographs as boundary conditions. This discussion is now provided in the revised manuscript (Discussion section, P.15 L9-12).*
* * *
Please comment on the larger spread produced by the WeGen approach.

*The weather generator tends to overestimate the spatial correlation of extreme precipitation. It is parameterized by using an isotropic correlation function by mixing low intensity large scale and high intensity local rainfalls. Hence, the generated fields of extreme precipitation tend to have larger spatial extent than observed. Please see the corresponding paragraph in the manuscript (P.12 L6 - 17)*

**Discussion:**

Weather generators in general: any weather generator makes a quite strong assumption about the tail behaviour, so that the higher the return period resp. the extremeness of the simulated precipitation, the larger should be the structural model uncertainty, which in turn is expected to quite influence the corresponding estimated hydrological load. While a 100 years event might be just at the boundary of what we might be able to extrapolate from about 40 years observations – with still some degree of confidence- anything beyond will very likely be strongly related to the tail models. Could you please elaborate on this, and state what do you think might be the impact of the use of another weather generator on your results?

*We agree that the extrapolation of weather generators make strong assumptions about the tail distribution and such the uncertainty raises alongside the exceedance probability which is directly propagated to the hydrological loads. The extrapolation beyond RPs of 100 years based on typically available data series of a few decades is associated with large uncertainties. Nonetheless, information about higher return periods are often required in practice (e.g. In Austria HQ300 is applied to define "residual risk areas" or the RP of 200 years is defined in the Solvency II definition in the European (re)insurance context (European Union (EU) 2009)). To our knowledge, there are no studies comparing risk assessments driven by different weather generators. Hence, it is difficult to make a reliable statement how decisive the tail dependence is with regards to the final risk estimates. On the one side, the effect of tail dependence is expected to increase with the return period. On the other side, the events with high return periods have low probability and might have little impact on the average risk (expected annual damage) (area under the risk curve). So, this is a question whether we look at the loss estimate of a e.g. 1000-year flood or we are interested in the annual expected damage. For the first, the tail dependence might be more important, for the second rather less important. We believe, more studies are needed to compare different weather generators and their impact on risk assessments.*

*The same framework with different components (e.g. weather generator or RR-model) likely lead to alternative results. The outcome of higher systematic risk estimates for the WeGen approach might not be true for other model components and thus should not be generalized. We added an additional paragraph in the Discussion section regarding this issue. (P14 L17-22)*
* * *
P13-L26: Actually in Figure 5 you are showing the "overall" uncertainties of the two modelling chains, what do you mean with and why do you write single uncertainty sources here?

*As stated on P.13-L20-21, the uncertainty presented in Figure 5 only shows the uncertainty which corresponds to the multiple realizations and does not account for other sources of uncertainties (e.g.*

*parameter uncertainties). Also, no comprehensive uncertainty assessment by propagating the uncertainties of all sub-models throughout the model chain is included in the current work, it is still possible to have a look at each individual modelling step. We reformulated the statement and the sentence about 'single uncertainty sources' was deled.*

This is just a consideration /suggestion: Of course volumes cannot be considered by applying the HT model, however besides flood peaks, flood volumes can play an important role in flood risk analysis. You correctly mention that one of the advantages of applying WeGen is the ability to produce continuous hydrographs (and accordingly event volumes), however you might want to mention it explicitly? Flood volumes play an important role for hydraulic infrastructure such as reservoirs/lakes/etc. (and thus in hydraulic design engineering), and also in the case of presence of floodplains with retention potential. On the other side, volumes might be another validation measure for the WeGen approach, as –depending also on how good is working the hydrological model- indirectly indicate how well or bad is the weather generator doing by reproducing persistence at longer time scales (a week and beyond), as I would generally expect this kind of weather generator to be underestimating persistence. This is something you might want to check in the future?

*Many thanks for this interesting suggestion. We agree that flood volumes are an important characteristic of flood events and especially relevant for risk assessment (e.g. Dung et al. 2015; Lamb et al. 2016). We will elaborate more on this in the discussion. Future research can focus on the ability of the model chain WeGen-RR to reproduce flood volumes. This is however not straightforward. This can be approached by the assessment of the areal precipitation volume for different durations and spatial aggregations (from small sub-catchments to the entire basin) and secondly, by comparing the observed and simulated flood event volume statistics similarly to the flood peak statistics. We added a notion about the relevance of flood volumes beside peak estimates to describe the severity of a flood event in the discussion section.*

**Technical corrections**

Figure 2: it is full of typing errors (refer to Obseravtions, topographie, Geometrie)

*The Figure was revised accordingly.*

Please use the word realizations instead of repetitions

*The term "realizations" is now used throughout the manuscript.*

Please use more consistently the word severity (e.g. in the of Figure 4 use return period instead of level of severity => what might be confusing, as you define and quantify severity by the UoFH later on)

*As suggested the caption was changed to return period to avoid confusion.*

P2-L3/4: what do you mean with floods hazard characteristics?

*For example, inundation depth or flow velocity. This information was added for clarification.*

P6-L24: "...a linear interpolated interpolation..." please reformulate better

*The term "interpolated" was deleted.*

P7-L8: ...can be statistically.

*corrected*

P10: please reformulate the last sentence (90% of exceeding sites sounds weird)

*The sentence will be revised to: "The most widespread event (UoFH=77) corresponds to about 90% of the sites exceeding the threshold."*

P11-L11: just a suggestion=> capability instead of feature?

*Thank you for the suggestion. We will use the term "capability" instead of feature.*

P11-L15/16: might effect ...?=> please reformulate

*'effect' was replaced by 'affect'*

P12-L5: …estimate of (=> better with? Or by?)) WeGen approach

*The manuscript was changed to "...estimate by "*

P12-L14: just a suggestion: instead of On the contrary => At the same time? On the other hand?

*Thank you for the suggestion. The phrase "At the same time" was used instead.*

P7-L21: dependence matrices instead of dependence metrices

P7-L26: Each simulation instead of Each simulations

P8-L1: the data are "too" few instead of the data are "to" few

P10-L6/7: either remove a , in "a significat lower damages.."or change damages to singular

P11-L1: …simulate (remove a) complex spatially heterogeneous patterns.

P11-L5: On the contrary. . .only indirectly …

P11-L13/14: One possible reason could ...be?

P12-L2: instead of overall estimation => overestimation?

*Thank you for the grammar and language corrections. The manuscript was revised accordingly.*

[revised manuscript text omitted]